# Anionic nanoplastic exposure induces endothelial leakiness

Wei Wei [1,2,10], Yuhuan Li [3,4,10], Myeongsang Lee [5],
Nicholas Andrikopoulos [4], Sijie Lin [6], Chunying Chen [7], David Tai Leong [8],
Feng Ding [5], Yang Song [1] ✉ & Pu Chun Ke [4,9] ✉

The global-scale production of plastics has been instrumental in advancing modern society, while the rising accumulation of plastics in landfills, oceans, and anything in between has become a major stressor on environmental sustainability, climate, and, potentially, human health. While mechanical and chemical forces of man and nature can eventually break down or recycle plastics, our understanding of the biological fingerprints of plastics, especially of nanoplastics, remains poor. Here we report on a phenomenon associated with the nanoplastic forms of anionic polystyrene and poly(methyl methacrylate), where their introduction disrupted the vascular endothelial cadherin junctions in a dose-dependent manner, as revealed by confocal fluorescence microscopy, signaling pathways, molecular dynamics simulations, as well as ex vivo and in vivo assays with animal model systems. Collectively, our results implicated nanoplastics-induced vasculature permeability as primarily biophysical-biochemical in nature, uncorrelated with cytotoxic events such as reactive oxygen species production, autophagy, and apoptosis. This uncovered route of paracellular transport has opened up vast avenues for investigating the behaviour and biological effects of nanoplastics, which may offer crucial insights for guiding innovations towards a sustainable plastics industry and environmental remediation.

Micro- and nano-plastics are derivatives of physical, chemical, and biological degradations of plastics discharged into the environment, from industrial and research outlets or as byproducts of commodities such as clothing, shampoos, and plastic tea bags[1]. With the global production of plastics steadily rising over the past century, reaching 359 million tons in 2018 alone[2], understanding and mitigating the adverse biological effects of these man-made materials[3], especially in their micro- and nanoplastic forms, has become crucial for the protection of human health and environmental sustainability while preserving the modern way of life.

Trace amounts of plastics have been found in animal and human organs via inhalation, ingestion, and dermal exposure, resulting from

[1]State Key Laboratory of Environmental Chemistry and Ecotoxicology, Research Center for Eco-Environmental Sciences, Chinese Academy of Sciences, Beijing 100085, China. [2]Key Laboratory of Luminescence Analysis and Molecular Sensing, Ministry of Education, College of Pharmaceutical Sciences, Southwest University, 2 Tiansheng Rd, Beibei District, Chongqing 400715, China. [3]Liver Cancer Institute, Zhongshan Hospital, Fudan University, Shanghai 200032, China. [4]Drug Delivery, Disposition and Dynamics, Monash Institute of Pharmaceutical Sciences, Monash University, 381 Royal Parade, Parkville, VIC 3052, Australia. [5]Department of Physics and Astronomy, Clemson University, Clemson, SC 29634, USA. [6]College of Environmental Science and Engineering, Tongji University, 1239 Siping Road, Shanghai 200092, China. [7]CAS Key Laboratory for Biomedical Effects of Nanomaterials and Nanosafety, National Center for Nanoscience and Technology of China, Beijing 100190, China. [8]Department of Chemical and Biomolecular Engineering, National University of Singapore, 4 Engineering Drive 4, Singapore 117585, Singapore. [9]Nanomedicine Center, The Great Bay Area National Institute for Nanotechnology Innovation, 136 Kaiyuan Avenue, Guangzhou 510700, China. [10]These authors contributed equally: Wei Wei, Yuhuan Li. ✉e-mail: yangsong@rcees.ac.cn; pu-chun.ke@monash.edu

their prevalence in air, water, and soil[4,5]. Differing from their micro-sized and more extensively studied counterparts in surface area per volume and toxicity profile, nanoplastics can impair animal growth, reproduction, and metabolism[6–9], and compromise the immune system by elevating cytokine secretion, apoptosis, endoplasmic reticulum stress, and oxidative stress[10–13].

Recently, it has been reported that anionic inorganic nanoparticles such as TiO$_2$, SiO$_2$, gold, and nanodiamonds in a given size range (i.e., <100 nm), can disrupt the vascular endothelial cadherin (VE-cadherin) junctions to create transient, micron-sized physical openings in endothelial monolayers[14,15]. This phenomenon, termed as nanomaterial-induced endothelial leakiness (NanoEL), has significant implications for nanotoxicology as well as nanomedicine, where nanostructures are designed to deliver drugs or detect tissue abnormalities via their vascular circulation[16], involving their translocation across the blood-brain barrier (BBB) at times. In this study, we report on a biological phenomenon associated with nanoplastics exposure, in the form of endothelial leakiness in human umbilical vein endothelial cells (HUVECs) elicited by the nanomaterials. This finding is surprising in that (1) nanoplastics are polymeric materials not known to induce NanoEL[16], and (2) the densities of the nanoplastics, i.e., ~1.05 g/m$^3$ for polystyrene and ~1.18 g/m$^3$ for poly(methyl methacrylate) (PMMA), are considerably lower than the threshold of 1.72 g/m$^3$ determined for NanoEL-competent inorganic nanoparticles[17]. Specifically, we examined the molecular mechanisms of VE-cadherin dimer rupture by polystyrene and PMMA nanoplastics using molecular dynamics simulations, and further documented endothelial leakiness ex vivo with rabbit and swine veins and in vivo with mice exposed to the nanoplastics. This study implicated vasculature permeability as a mechanism for the paracellular transport of nanoplastics, filling a crucial knowledge void in our quest of understanding the biological behaviour and fate of plastics crowding the ecosphere.

## Results and discussion

### Characterisations of polystyrene nanoplastic and their cytotoxicity

The morphology and hydrodynamic diameter of the polystyrene (PS) nanoplastic were characterised by transmission electron microscopy (TEM) and dynamic light scattering (DLS). As displayed in Fig. 1a, b, and Supplementary Fig. 1, the PS beads were monodispersed and assumed an average size of $21.2 \pm 3.5$ nm in water and $26.2 \pm 4.6$ nm in endothelial cell medium (ECM) as indicated by TEM. In comparison, the PS nanoplastic displayed the hydrodynamic size of $62 \pm 5.0$ nm in H$_2$O and $72 \pm 2.0$ nm in ECM as determined by DLS, due to some degree of agglomeration. Accordingly, the ζ-potential of the nanoplastic changed from $-35.4 \pm 1.9$ mV in H$_2$O to $-7.5 \pm 0.5$ mV in ECM (Supplementary Table 1). In addition, X-ray photoelectron spectroscopy (XPS) analysis indicated the presence of carboxyls on the nanoplastic (Fig. 1c). The toxicity of the nanoplastic was further quantified by an in vitro Cell Counting Kit-8 (CCK 8), reactive oxygen species (ROS) and cellular mortality assays, where HUVECs were exposed to different concentrations of the nanoplastic over time (Fig. 1d, e and Supplementary Fig. 2). HUVECs with compromised DNA integrity and/or membranes were indicated by propidium iodide (PI) staining with a 22 h treatment (Fig. 1d and Supplementary Fig. 2a), and no obvious cell death occurred for incubation up to 10 h for all nanoplastic concentrations applied. Shrinkage and deformation of the cells occurred with PS nanoplastic of 0.5 mg/mL after 22 h, while no significant changes in either cell viability or membrane damage were observed at PS concentrations of 0.05 to 0.25 mg/mL. For shorter periods of treatment, the cell viability was not affected by PS nanoplastic at 0.05 and 0.5 mg/mL within 1 h of exposure (Fig. 1e), however, ROS production was positively correlated with the nanoplastic concentration and time. Specifically, ROS were observed at low nanoplastic

concentrations (0.05 and 0.1 mg/mL) over 1 h (Supplementary Fig. 2b, c). Internalisation of PS nanoplastic in HUVECs was enhanced with increased incubation time and dose (Fig. 1f and Supplementary Fig. 3). Furthermore, we studied the way PS nanoplastic incited cell death. Autophagy is a self-digestion process[18] involved in both cell death and survival[19]. The lipidated form of microtubule-associated protein light chain 3 (LC3), LC3-II has been shown to be an autophagosome marker, and the formation or turnover rate of LC3-II is used as an indicator of autophagy activity[20]. We found that 0.05 mg/mL PS nanoplastic did not increase conversion of LC3-II in HUVECs at 1 and 3 h, but significantly up-regulated the expression level of LC3-II at 6 h. Moreover, abundant expression of LC3-II could be detected in just 1 h in the presence of 0.5 mg/mL PS nanoplastic (Fig. 1g). In addition, PS nanoplastic induced autophagy and apoptosis by inhibiting the PI3K/AKT pathway in HUVECs at 6 h (Supplementary Figs. 4 and 5). Specifically, western blots showed that autophagy-related proteins Beclin1, p62, and Atg5 and pro-apoptotic protein Bax were significantly increased, while anti-apoptotic protein Bcl-2 was significantly decreased in HUVECs in a dose-dependent manner. The results of gene expression and protein expression were in good agreement (Supplementary Figs. 4 and 5). When HUVECs were exposed to 0.05 or 0.5 mg/mL PS nanoplastic for 6 h, atrophic nuclei and formation of apoptotic bodies were observed (Supplementary Fig. 6).

### Polystyrene nanoplastic-induced endothelial leakiness in vitro

Anionic inorganic nanoparticles can migrate to and disrupt the adherens junctions between endothelial cells to induce NanoEL[21]. Interestingly, we observed NanoEL in HUVECs in the presence of polymeric PS nanoplastic. We postulated that nanoplastic could disrupt the adherens junctions and further cause endothelial leakiness (Fig. 2a). To confirm the occurrence of endothelial leakiness, we performed a transwell assay with a fluorescent probe, fluorescein isothiocyanate conjugated dextran (FITC–dextran), with an intact HUVECs monolayer exposed to various concentrations of PS nanoplastic (Fig. 2b). A significant leakiness was observed when the cells were treated with 0.5 mg/mL of the nanoplastic within 1 h of exposure. As the treatment increased to 6 h, we observed the appearance of endothelial leakiness for all concentrations of the nanoplastic. We then employed immunofluorescence staining to visualise disruption of adherens junctions in the endothelial barriers, where two representative concentrations of PS nanoplastic at 0.05 and 0.5 mg/mL were applied to HUVECs, and 0.5 mM of EDTA was used as positive control (Fig. 2c, Supplementary Figs. 7 and 8). Abundant gaps were observed for 0.05 and 0.5 mg/mL of the nanoplastic at 6 h. Morphological changes and integrity disruption to VE-cadherin in HUVECs were noticeable at 3 and 6 h compared with control. The extent of endothelial leakiness was determined through a gap area analysis using ImageJ (Fig. 2d and Supplementary Fig. 7)[21]. The derived percentages of gap areas corroborated data from the transwell assay, where endothelial leakiness was more pronounced over time for PS nanoplastic of 0.05 and 0.5 mg/mL. The extent of endothelial leakiness at 6 h revealed a notable elevation, compared with 1 h and 3 h incubations. Concomitantly, the fluorescence intensity alteration of actin filaments indicated reorganisation of the cytoskeletal actin network, in conjunction with the endothelial leakiness phenomenon, at 3 and 6 h exposure of the nanoplastic at 0.05 mg/mL and 1, 3, 6 h exposure of the nanoplastic at 0.5 mg/mL, respectively (Fig. 2e). In addition, a VE-cadherin protein pull-down assay was implemented to investigate whether PS nanoplastic could directly bind and disrupt the adherens junctions. After exposure to different concentrations of PS nanoplastic and for different durations, the whole cells were lysed and PS nanoplastic with their associated proteins in culture media were pulled down by centrifugation at the same time. As shown in Fig. 2f and Supplementary Fig. 9, the homophilic VE-cadherin of the adherens junctions was pulled down by PS nanoplastic in a dose-dependent manner, and an elevation of VE-cadherins bound to PS was observed at

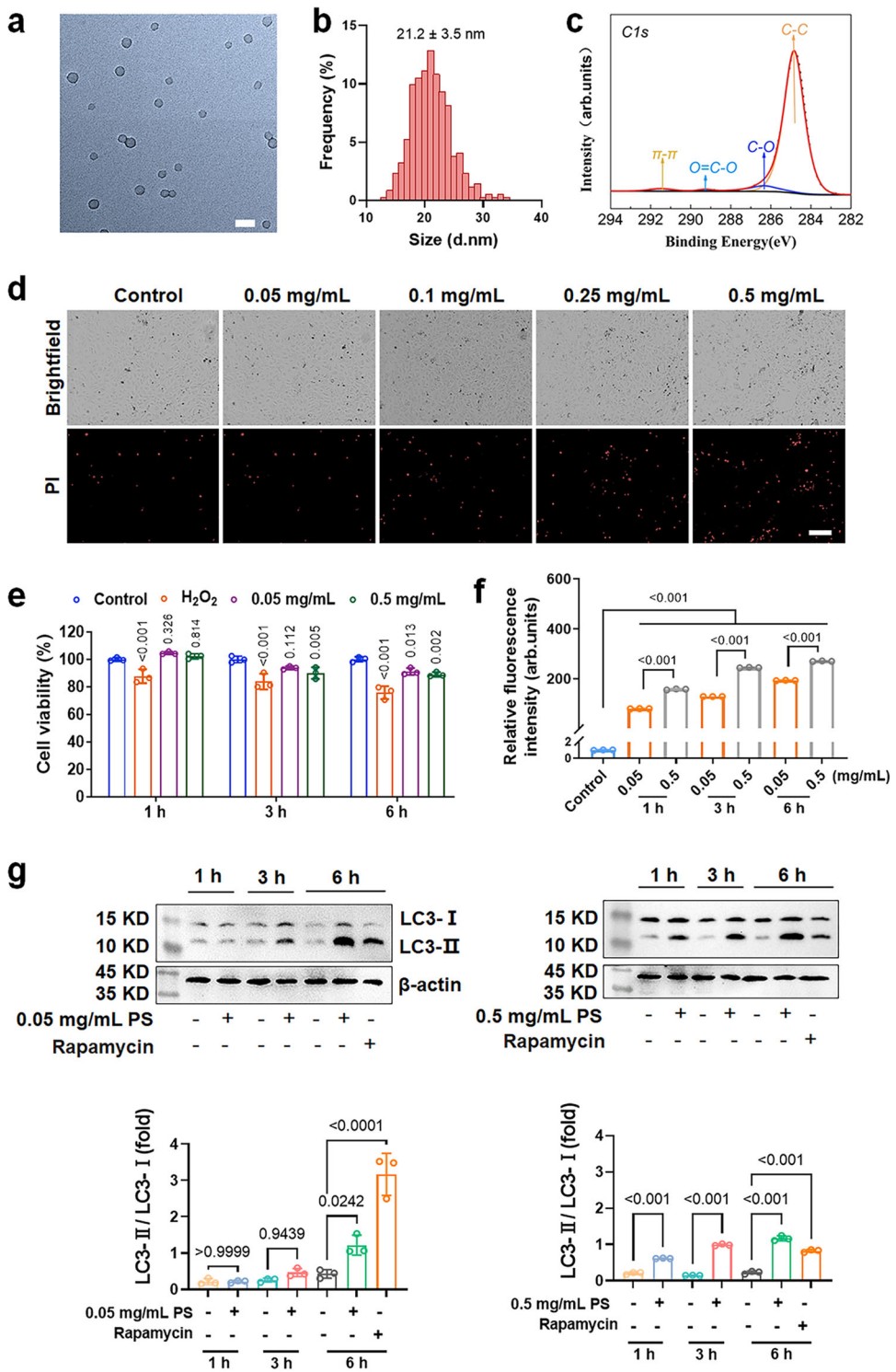

0.5 mg/mL than 0.05 mg/mL, whereas the amount of VE-cadherins pulled down by PS nanoplastic was not influenced by the exposure duration. Additionally, PS nanoplastic of 0.05 and 0.5 mg/mL were added to a non-treated cell lysate (post lysis, P), which did not pull down any detectable VE-cadherins. Interpretation of the specific result showed PS nanoplastic bound to VE–cadherins within the adherens junctions and not in the lysis buffer condition, suggesting that an intact adherens junction was necessary for PS nanoplastic to bind to VE-cadherin. On the other hand, there were no pull-down VE-cadherins in the presence of protein A/G magnetic beads (A) after cell lysis without PS nanoplastic. Whole cell lysate showed a similar expression of VE-cadherin across the various groups, indicating that PS nanoplastic did not affect cellular VE-cadherin expression transcriptionally or post-translationally.

## Characterisations of other types of nanoplastics and their NanoEL competence

Apart from the carboxylated PS nanoplastic as described above, we also employed two other types of nanoplastics, namely, aminated polystyrene (NH₂-PS) and poly(methyl methacrylate) (PMMA), to explore the NanoEL competence of nanoplastics of different charge and chemical composition.

**Fig. 1 | Characterisations of polystyrene nanoplastic and their cytotoxicity.**
**a, b** TEM imaging of polystyrene (PS) nanoplastic in Milli-Q water and their corresponding size distribution ($n = 341$ nanoparticles examined). Scale bar: 50 nm.
**c** Functional groups of PS nanoplastic were determined by X-ray photoelectron spectroscopy (XPS) analysis. The dominant peak at 284.83 eV arose from C–C bonding, and the other peaks at 286.34 eV and 289.28 eV were attributed to C–O, and O=C–O bonding, respectively. **d** Toxicities of HUVECs upon exposure to different concentrations of PS nanoplastic at 22 h. Dead cells stained with PI were revealed in the red fluorescence channel ($n = 3$ biological replicates). Scale bars: 200 μm. **e** Cell viability with PS nanoplastic at 0.05 and 0.5 mg/mL was measured by a CCK8 assay at different times (1, 3, and 6 h). $H_2O_2$ (200 μM) was used as positive control. Statistical analysis was performed through a one-way ANOVA followed by Tukey's multiple comparison tests. $P$ values comparing to control in each group were inserted in the panel. **f** Evaluation of cell association for different concentrations of PS nanoplastic at 1, 3, and 6 h. The green fluorescence from nanoplastic was measured by flow cytometry. Data are expressed as means ± SD ($n = 3$). Statistical analysis was performed through a two-way ANOVA followed by Tukey's multiple comparison tests. The derived $P$ values were inserted in the panel. **g** Western blot and semi-quantitative analysis of microtubule-associated protein light chain 3-II/I (LC3-II/I) expression level. HUVECs were exposed to PS nanoplastic (0.05 and 0.5 mg/mL) for different times (1, 3 and 6 h). Rapamycin (1 μM) with 6 h-treatment was used as positive control. Protein levels were standardised by comparison with β-actin. Data are expressed as means ± SD. Biologically independent samples were used ($n = 3$). Statistical analysis was performed through two-way ANOVA followed by Tukey's multiple comparison tests. The derived $P$ values were inserted in the panel. Source data are provided as a Source Data file.

Similarly, both $NH_2$-PS and PMMA nanoplastics were characterised by TEM, DLS and XPS. In particular, TEM revealed the average size of $NH_2$-PS and PMMA as $19.7 ± 3.8$ nm and $54.7 ± 6.1$ nm, respectively (Fig. 3a, Supplementary Figs. 10 and 11), while DLS measurements indicated two peaks of the hydrodynamic size of $31 ± 0.6$ nm and $3984 ± 253.4$ nm for $NH_2$-PS nanoplastic and $89 ± 1.3$ nm for PMMA nanoplastic in $H_2O$ (Supplementary Table 1). Furthermore, ζ-potential measurements revealed a positive surface charge of $31.0 ± 1.3$ mV for $NH_2$-PS nanoplastic and a negative surface charge of $-24.8 ± 0.7$ mV for PMMA (Supplementary Table 1). The size and ζ-potential of nanoplastic were also tested in ECM by DLS, which did not show significant changes. The surface characterisations of both nanoplasitcs were further analysed with XPS, revealing the presence of amidogens on the $NH_2$-PS nanoplastic and carboxyls on the PMMA nanoplastic as expected (Fig. 3b).

Moreover, we exposed HUVECs to both types of nanoplastics for different concentrations and durations of treatment. In particular, 1 h of HUVECs exposure to $NH_2$-PS nanoplastic at 0.05 mg/mL did not affect cell viability, while cell mortality was significantly increased with the increasing dosage and incubation time of the nanoplastic (Fig. 3c). In comparison, the negatively charged PMMA nanoplastic displayed better biocompatibility and innocuity (Fig. 3d). Furthermore, to observe disruption to VE-cadherin junctions and quantify NanoEL entailed by nanoplastics, immunofluorescence staining of VE-cadherins was employed and permeability of the HUVECs endothelial monolayer was measured by a transwell assay (Fig. 3e-g and Supplementary Fig. 12). Specifically, PMMA nanoplastic already exhibited a capability of inducing NanoEL at the low concentration of 0.05 mg/mL in 1 h, which became more pronounced with increased nanoplastic concentration and exposure time. Not surprisingly, $NH_2$-PS nanoplastic exerted little impact on endothelial leakiness. Overall, these results indicated that NanoEL could be triggered by negatively charged nanoplastics of 20–60 nm in size, for both PS and PMMA.

ROS production is a hallmark of cytotoxicity[22]. Many metal oxide nanoparticles can induce cellular ROS[23], indirectly leading to endothelial leakiness through cytoskeletal damage[24]. On the other hand, cellular uptake is not necessary for the occurrence of the leakiness[21]. Among the relevant biomarkers, 4-amino-5-(4-methylphenyl)−7-(t-butyl)pyrazolo[3,4-d]-pyrimidine (PP1) is a Src kinase inhibitor that inhibits VE-cadherin phosphorylation[25], rho-associated protein kinase (ROCK) inhibitor Y-27632 prevents perturbation to the cytoskeletal network[26], and N-Acetyl-L-cysteine (NAC) inhibits ROS generation. We found that endocytosis inhibitors methyl-β-cyclodextrin (MβCD) and monodansylcadaverine (MDC) could not prevent the leakiness induced by the PS nanoplastic, while PP1 and Y-27632 could stall the process (Fig. 4a). We observed a significant increase in ROS level after HUVECs were treated with PS nanoplastic for 1 h (Supplementary Fig. 2). There was no significant difference in the level of leakiness when the cells were treated with the antioxidant NAC (5 mM) or endocytosis inhibitors MβCD (5 mM) and MDC (10 μM) for 1 h prior to nanoplastic exposure (0.05 mg/mL) for 1 h

(Fig. 4b). However, compared with the PS group, leakiness levels in the PP1 group and the Y-27632 group were significantly decreased ($P < 0.05$ or $P < 0.001$). Consistent results were obtained by treating cells with 0.5 mg/mL PS nanoplastic (Fig. 4c). The above phenomenon indicated that PS nanoplastic-induced endothelial leakiness was independent of ROS formation and endocytosis but was related to VE-cadherin pathway and actin remodeling.

The natural state of endothelial cells in mature blood vessels is a fused and static monolayer, in which VE-cadherins are gathered at the contacts between cells, and adhere to the corresponding tissues[27]. VE-cadherin contributes to cytoskeleton rearrangement[28] and tight junctions remodeling[29] in endothelial cells during angiogenesis. Although it has other functions, the main function of VE-cadherin is to promote adhesion between endothelial cells and to increase cell permeability when junctions are disrupted[27]. The tyrosine kinase Src activates phosphorylation of VE–cadherin at the Y658 and Y731 residues[30]. VE-cadherin interacts with p120 and β-catenin through its cytoplasmic tail to promote cell adhesion[31]. Tyrosine phosphorylation of the cytoplasmic tail of VE-cadherin disrupts p120 binding and destabilises the interaction between VE-cadherin and cytoplasmic catenin, thereby forming gaps through the lateral displacement of phosphorylated VE-cadherin[32]. We used PP1 to detect whether the PS nanoplastic activated Src kinase to induce VE-cadherin phosphorylation. For HUVECs treated with PS nanoplastic of 0.05 or 0.5 mg/mL, PP1 prevented VE-cadherin phosphorylation at the Y658 and Y731 residues (Fig. 4d, e). There were significant decreases in the phosphorylation levels of the Y658 and Y731 residues of VE-cadherin when the cells were treated with PP1 (10 μM) for 1 h prior to PS nanoplastic introduction (0.05 or 0.5 mg/mL) for various times (1, 3 or 6 h) (Fig. 4f). Our data further showed that PS nanoplastic can trigger the phosphorylation of VE-cadherin, which may increase intercellular gaps and induce leakiness. In addition, we found that PMMA (0.05 or 0.5 mg/mL) also induced phosphorylation of VE-cadherins (Supplementary Fig. 13).

## Molecular dynamics simulations of nanoplastic binding with VE-cadherin dimers

To characterise VE-cadherin dimer disruption induced by PS nanoplastic, we performed all-atom discrete molecular dynamics (DMD) simulations with implicit solvent models. Before exerting an in silico pulling force, binding simulation of a cadherin dimer with PS nanoplastic was first performed. We employed the first extracellular domain (EC1) dimer of full-length VE cadherin, since the domain-swapped region in the EC1 dimer is important for *trans* cell-to-cell adhesion[21] (Fig. 5a). Next, PS nanoplastic with carboxyl surface groups were constructed to investigate their competence with NanoEL. Due to the high computational cost related to the large size of PS, we modeled PS nanoplastic composed of 20 repetitive PS chains and the diameter of the formed nanoplastic was equivalent to ~15 Å for the 50 ns equilibrium DMD simulation (Fig. 5b). Non-charged PS chains of similar sizes have been shown both computationally and experimentally to directly interact with cells and penetrate plasma membranes[33,34]. Here,

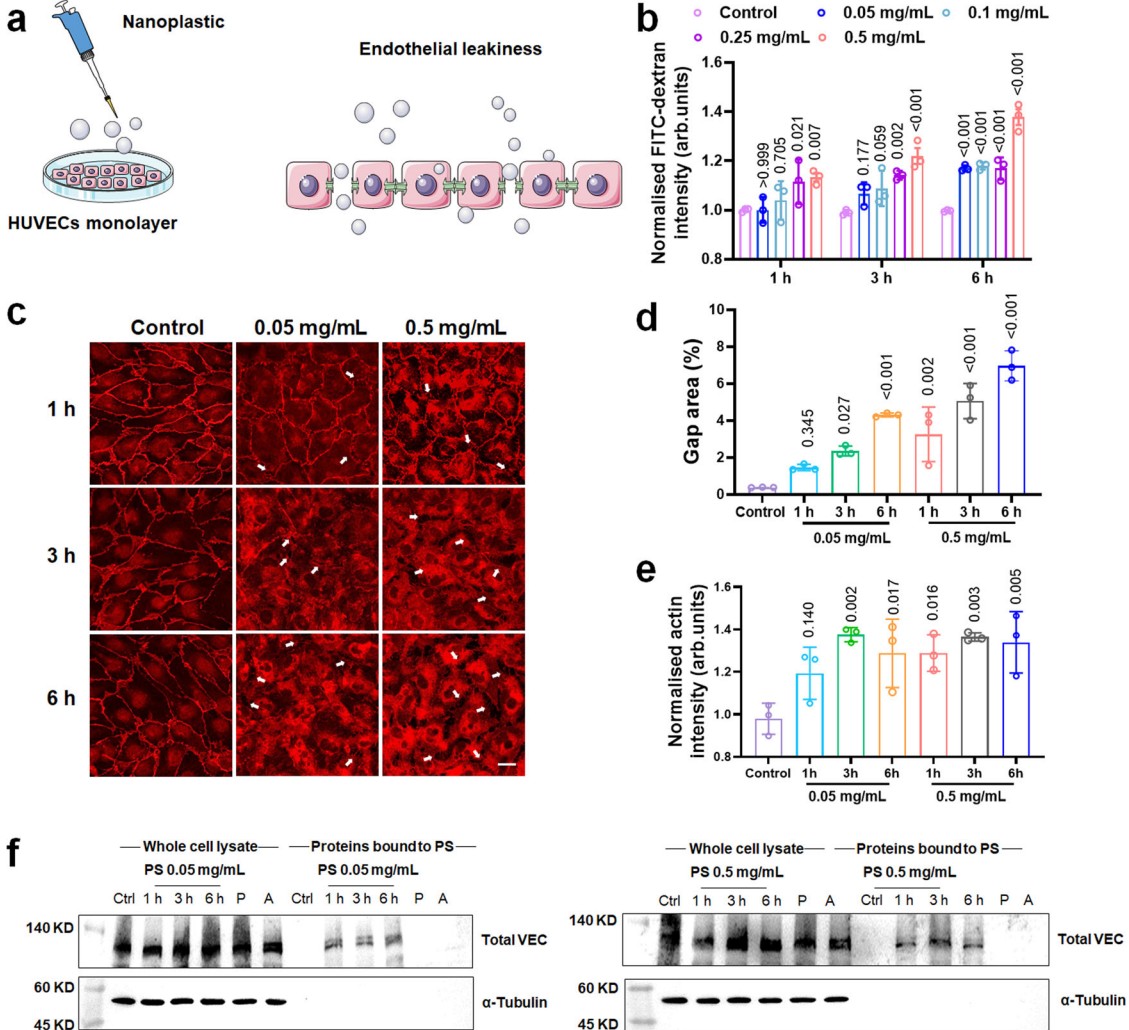

**Fig. 2 | Polystyrene nanoplastic-induced endothelial leakiness in HUVECs.**
**a** Illustrated interaction between adherens junctions and polystyrene (PS) nanoplastic. The integrity of the HUVECs monolayer was disrupted by the PS nanoplastic due to their interaction with VE-cadherin. **b** Transwell assay indicated a dose- and time-dependence in the leakiness of endothelial cell barriers. Data are expressed as means ± SD (*n* = 3 biologically independent samples). Statistical analysis was performed through one-way ANOVA followed by Tukey's multiple comparison tests. The derived *P* values compared to control in each group were inserted in the panel. **c** Confocal fluorescence microscopy observed endothelial leakiness in the presence of different concentrations of PS nanoplastic (0.05 and 0.5 mg/mL) upon 1, 3, and 6 h exposure (*n* = 3 biological replicates). VE-cadherins were stained with red color. The white arrows indicate PS-induced gaps between HUVECs. Scale bar: 20 µm. **d** Semi-quantitative analysis of gaps area was performed by ImageJ software according to the images from panel **c**. Data are expressed as means ± SD (*n* = 3 biologically independent experiments). Statistical analysis was performed through

two-way ANOVA followed by Tukey's multiple comparison tests. The derived *P* values comparing to control were inserted in the panel. **e** Actin intensity was analysed by ImageJ software corresponding to the images in Supplementary Fig. 7. Actin filaments were stained by phalloidin-iFluor 488. Data are expressed as means ± SD (*n* = 3 biologically independent experiments). Statistical analysis was performed through two-way ANOVA followed by Tukey's multiple comparison tests. The derived *P* values comparing to control were inserted in the panel. **f** PS directly bound to adherens junctional homophilic VE-cadherins (VEC) in a dose-dependent manner. Post lysis (P), addition of PS at 0.05 and 0.5 mg/mL to a non-treated control did not pull down any detectable VE-cadherins. Protein A/G magnetic beads (A) were added to show that no detectable VE-cadherins were precipitated without the addition of PS (*n* = 3 biologically independent experiments). Source data are provided as a Source Data file. (Some art elements in panel **a** are from smart.servier.com.).

negatively charged PS modified with carboxyl groups was modeled as in the experiments to minimise their direct interactions with cell membranes containing negatively charged lipids. Subsequently, the constructed nanoplastic was randomly located from the EC1 dimer and 30 independent binding simulations were carried out to identify the binding sites of PS nanoplastic with the EC1 cadherin dimer. The computed binding frequency indicated that, after 50 ns, PS nanoplastic strongly bound to residues 32–38, 42, 46–51, and 76–81 in the VE-cadherin dimer (Supplementary Fig. 14a). These regions were located away from the interfacial region of the dimer and were rich in positively charged residues, as indicated by the surface representation of the dimer color-coded according to the nanoplastic-binding

frequencies (Fig. 5c). The binding frequency data suggested that the PS nanoplastic preferred to bind near the turn region of the dimer. Unmodified PS chain, on the other hand, bound to the dimer interface (residues 1–4, 85–89) due to hydrophobic interactions (Supplementary Fig. 15a, b). Therefore, the electrostatic interactions between the carboxyl groups on the surface of the PS nanoplastic and the positively charged residues in the turn region of the dimer led to association of the two entities.

In addition to the binding simulation, we further carried out a steered DMD (sDMD) simulation to determine the effects of PS nanoplastic on cadherin dimer stability as well as cadherin dissociation. Here, we considered a low force range (0–20 pN), which was sufficient

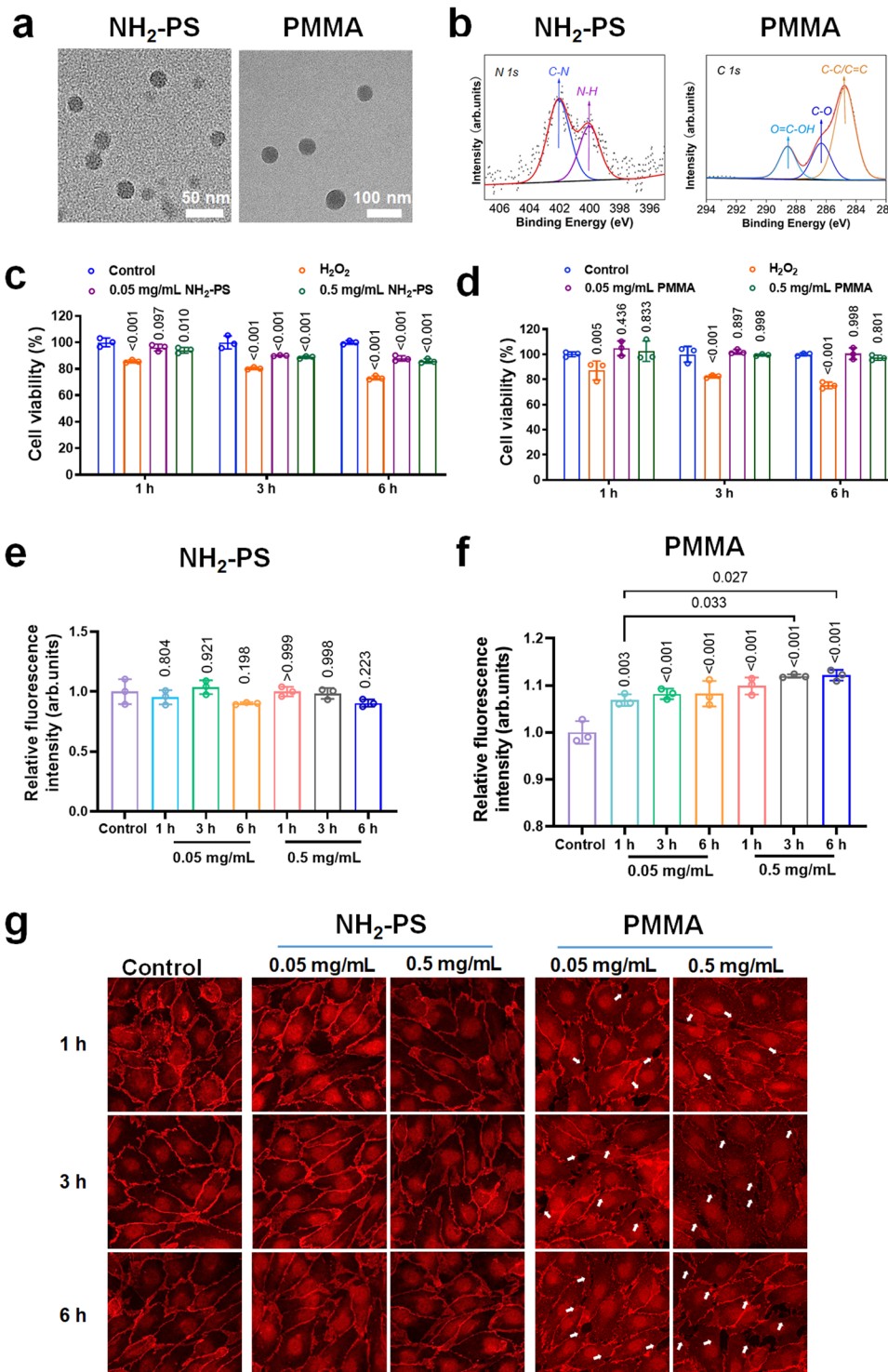

**Fig. 3 | Characterisations of NH2-PS and PMMA nanoplastics and their NanoEL competence with HUVECs. a** TEM imaging of NH$_2$-PS and PMMA nanoplastics in H$_2$O ($n$ = 3 independent samples). **b** Functional groups of NH$_2$-PS and PMMA nanoplastics were determined by XPS analysis. The major peaks for NH$_2$-PS at 402.06 and 400.02 eV were referable to C–N and N–H bonding, respectively. The dominant peak for PMMA at 284.74 eV arose from C–C or C=C bonding, and the peaks at 286.33 eV and 288.57 eV can be attributed to C–O, and O=C–OH bonding, respectively. **c**, **d** Cell viability of NH$_2$-PS and PMMA nanoplastics at 0.05 and 0.5 mg/mL, measured by a CCK8 assay at different times (1, 3, and 6 h). H$_2$O$_2$ (200 μM) was used as positive control. Data are expressed as means ± SD ($n$ = 3 biologically independent experiments). Statistical analysis was performed through one-way ANOVA followed by Tukey's multiple comparison tests. The derived $P$ values comparing to control were inserted in the panel. **e**, **f** Transwell assay indicated positive-charged NH$_2$-PS nanoplastic was incompetent in inducing endothelial leakiness, while negatively charged PMMA nanoplastic triggered leakage of endothelial cell barriers. Data are expressed as means ± SD ($n$ = 3 biologically independent experiments). Statistical analysis was performed through two-way ANOVA followed by Tukey's multiple comparison tests. The derived $P$ values comparing to control were inserted in the panel. **g** Confocal fluorescence microscopy revealed endothelial leakiness in the presence of PMMA nanoplastic (0.05 and 0.5 mg/mL) upon 1, 3, and 6 h treatments ($n$ = 3 biologically independent experiments). The white arrows indicate gaps between HUVECs. While no endothelial leakiness was observed in NH$_2$-PS nanoplastic treatment. Scale bar: 20 μm. Source data are provided as a Source Data file.

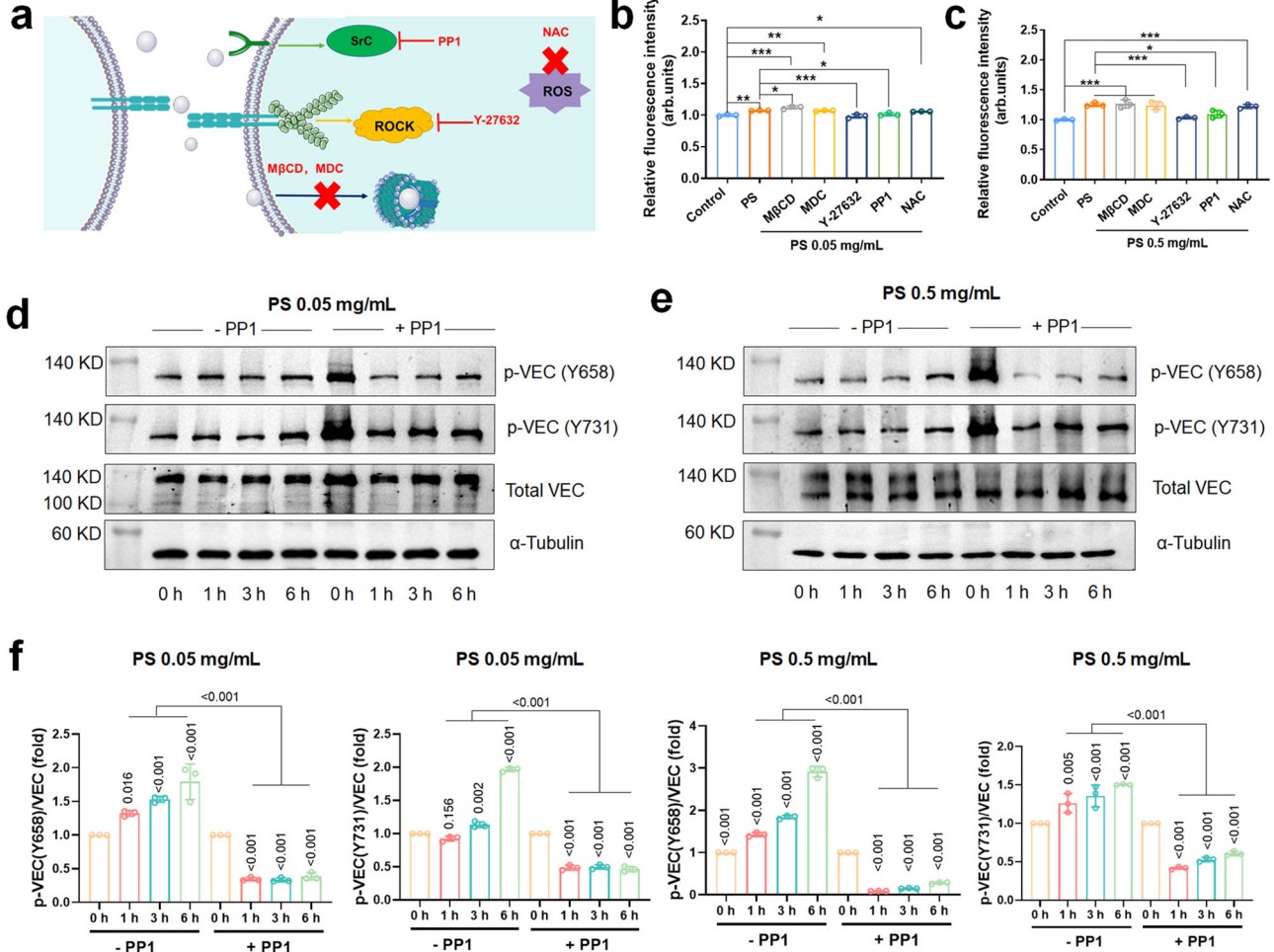

**Fig. 4 | Polystyrene nanoplastic-induced endothelial leakiness is independent of ROS formation and endocytosis but is related to VE-cadherin signaling pathway and actin remodeling. a** Blocking endocytosis with inhibitors or ROS inhibition did not prevent leakiness of PS nanoplastic. However, Src kinase inhibitor PP1 and ROCK inhibitor Y-27632 affected the leakiness of PS nanoplastic. HUVECs were treated with Src kinase inhibitor PP1 (10 μM), rho-associated protein kinase (ROCK) inhibitor Y-27632 (10 μM), endocytosis inhibitors (5 mM methyl-β-cyclodextrin (MβCD) and 10 μM monodansylcadaverine (MDC)), or ROS inhibitor (5 mM N-Acetyl-L-cysteine (NAC)) for 1 h prior to **b** 0.05 mg/mL or **c** 0.5 mg/mL PS nanoplastic treatment. PP1 and Y-27632 significantly reduced the fold of FITC-dextran penetration compared to their respective counterparts without inhibitor treatments. MβCD, MDC, or NAC did not significantly decrease the fold of FITC-dextran penetration compared to their respective counterparts without inhibitor

treatments. Data are expressed as means ± SD. Biologically independent samples were used ($n = 3$). Statistical analysis was performed through one-way ANOVA followed by Tukey's multiple comparison tests. The derived $P$ values were inserted in the panel. Western blot analysis of VE-cadherin and its phosphorylation levels: **d** 0.05 mg/mL or **e** 0.5 mg/mL PS nanoplastic treatment induced tyrosine phosphorylation of VE-cadherin at Y658 and Y731. However, PP1 effectively inhibited PS nanoplastic-induced phosphorylation of VE-cadherin at Y658 and Y731. **f** Semi-quantitative analysis revealed activation of VE-cadherin (VEC) signaling exposed to PS nanoplastic (0.05 or 0.5 mg/mL). Data are expressed as means ± SD ($n = 3$ biologically independent experiments). Statistical analysis was performed through two-way ANOVA followed by Tukey's multiple comparison tests. The derived $P$ values were inserted in the panel. Source data are provided as a Source Data file.

for rupturing the EC1 dimer by gold nanoparticles[21], to measure the cadherin dimer stability. We immobilised one of the EC1 domains and applied a constant force to the other EC1 domain with the direction pointing towards the associated EC2 domain (Supplementary Fig. 14b). For each case of forces applied, 30 independent sDMD simulations with initially randomised velocities were carried out for 100 ns. For the sDMD simulations, we computed the first mean dissociation time of the cadherin dimer when the number of atomic contacts between the cadherin dimer were reduced to zero. A violin plot was obtained as a function of dissociation time and applied constant forces (Fig. 5d). We found out that the PS nanoplastic induced high probabilities of early EC1 dimer dissociation compared to control of cadherins without PS nanoplastic. High probabilities of short dissociation times were observed as the magnitude of the applied forces was increased. Representative trajectories revealed that bound nanoplastic caused early dissociation of the cadherin dimer (Fig. 5e). In contrast, binding

of unmodified PS chain to the dimer interface did not destabilise the dimer (Supplementary Fig. 15c). To confirm the reduced stability of the cadherin dimer induced by PS nanoplastic, we further computed the distribution of dimer angle and root mean square fluctuation (RMSF) of the flexible domains of the dimer for the first 30 ns of the simulation without applied forces (Supplementary Fig. 14c, e). We previously found that inorganic gold nanoparticles shifted the EC1 dimer from stable (s-dimer) to intermediate states (x-dimer) while reduced the entropy of the cadherin dimer to disrupt its inherent function[21]. In the present work, our computational results suggested that the binding of PS nanoplastic destabilised the cadherin dimer via a similar molecular mechanism by altering the dimer state from s-dimer to x-dimer and reducing the flexibility of the dimer.

In addition to PS, we also investigated how PMMA nanoplastic might induce disruption to the cadherin dimer. The negatively charged 20mer PMMA modified by removing methyl groups of a subset of

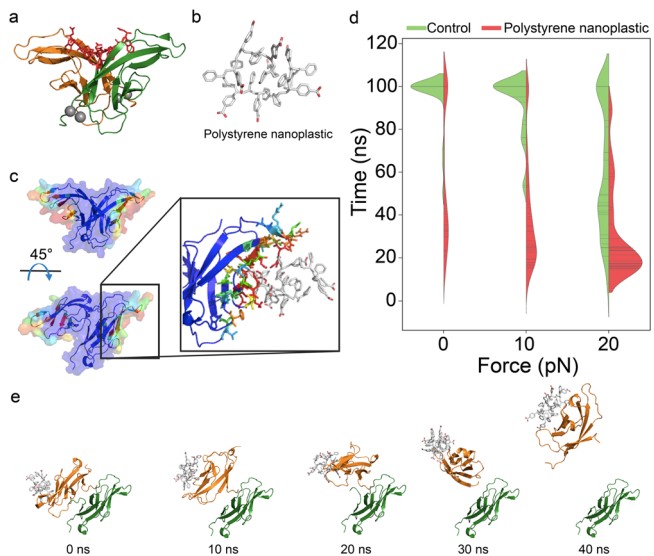

**Fig. 5 | Discrete molecular dynamics (DMD) and steered DMD (sDMD) simulations of the binding frequency of polystyrene (PS) nanoplastic with a VE-cadherin dimer as well as cadherin dimer stability. a** Structure of the EC1 cadherin dimer. Red sticks and gray spheres indicate the domain-swapped region and calcium ions, respectively. **b** Structure of a carboxylated PS nanoplastic after 50 ns equilibrium DMD simulation. **c** Colour-coded binding frequency of the PS nanoplastic on the EC1 cadherin dimer surface. Blue and red colors represent low to high binding frequencies. The enlarged panel details nanoplastic binding with the EC1 dimer. **d** Violin plots of the dimer binding with the PS nanoplastic. **e** Representative dissociation trajectories of the dimer in the presence of the PS nanoplastic under 0 pN of pulling.

composite repeats was found to form a compact globule after 50 ns equilibrium DMD simulations (Supplementary Fig. 16a). We calculated the binding frequency of PMMA with the cadherin dimer via 50 ns binding DMD simulations. PMMA nanoplastic had a similar strong binding to the same turn regions in cadherin as did PS nanopastics (Fig. 5a), but also a weak binding to the interface regions of the dimer (Supplementary Fig. 16b, c). Subsequent sDMD simulations of the cadherin dimer with PMMA nanoplastic indicated that PMMA nanoplastic also increased cadherin dimer dissociation under low forces (Supplementary Fig. 16d). Interestingly, due to the additional binding of PMMA to the cadherin dimer interface, an alternative dissociation pathway with competitive binding of the nanoplastic to one of the cadherin monomers was observed (Supplementary Fig. 16e). Regardless of their chemical composition, both anionic PS and PMMA nanoplastics induced dissociation of the cadherin dimer. Although the sizes of the PS and PMMA nanoplastics were not directly comparable to those in the experiments, due to the need for reducing the computational costs, our additional simulations showed that a range of PS nanoplastics with increased size up to 80 mer consistently disrupted the cadherin dimer (Fig. 5e and Supplementary Fig. 17). Hence, large nanoplastics are expected to behave similarly in interacting with VE-cadherin dimer. Taken together, our in silico data corroborated the experimental observations that PS and PMMA nanoplastics were competent in disrupting cadherin-cadherin association and our simulations further revealed that nanoplastic binding reduced the cadherin dimer stability through increased inter-domain strain and a reduced entropy.

### Polystyrene nanoplastic-induced endothelial leakiness in vessels ex vivo

In addition to the in vitro and in silico examinations, we also performed ex vivo assays using rabbit and swine vessels to measure leakiness in these vascular endothelia induced by PS nanoplastic (Fig. 6a). It was observed that the fluorescence intensities of leaked EBD in rabbit vessels were elevated with the increased concentration of PS nanoplastic (Fig. 6b). We observed a similar phenomenon with swine vessels (Fig. 6c). When swine vessels were treated with 0.5 and 5 mg/mL PS nanoplastic for 6 h, the leakiness of swine vessels was significantly enhanced compared with control ($P < 0.001$). At the same time, we also detected phosphorylation of VE-cadherins ex vivo (Supplementary Fig. 18).

### Polystyrene nanoplastic-induced endothelial leakiness in vivo

We further studied endothelial leakiness induced by PS nanoplastic in vivo. For studies using intravenous delivery of nanoparticles, the nanoparticles mainly accumulate in the liver and spleen, etc.[35,36]. In our experiment, mice were subjected to intravenous injection of EBD solution containing PS nanoplastic. After 24 h, we observed that, for PS nanoplastic at doses of 1.5 mg/kg, 15 mg/kg, and 30 mg/kg, the leakiness of EBD mainly existed in the brain, liver, spleen, and lungs of the mice (Fig. 6d). There was also a small amount of fluorescence of leaked EBD in the kidneys and diaphragms. We quantitatively analysed the fluorescence intensities of leaked EBD in tissues. Except for the heart, the fluorescence intensities of leaked EBD in the brain, liver, spleen, lungs, kidneys, and diaphragms increased significantly with the introduction of the PS nanoplastic (Supplementary Fig. 19). Endothelial cells provide a physical barrier between the circulation and the underlying tissue. However, certain stimuli can cause partial disruption to the endothelial barrier, thereby increasing fluid extravasation from the circulation to the interstitium above basal levels[37]. Long-term increase in permeability can destruct cell-cell junctions, which in turn results in altered vascular integrity[27]. To investigated whether PS nanoplastic could cause leakiness in subcutaneous blood vessels, we injected EBD into the tail veins of mice, followed by either vehicle control PBS buffer or PS nanoplastic injected into the subcutaneous pockets. PS nanoplastic caused EBD extravasation at the subcutaneous vasculature based on quantifications, thereby confirming the occurrence of endothelial leakiness (Fig. 6e).

Understanding the environmental burden including the lifecycle and carbon footprint of fossil plastics is a grand challenge facing the world today. This challenge is imposed by the ever-increasing presence of industrial-scale plastics production coupled with their remarkably gradual degradation in the natural environment. Indeed, the potential adverse effects of plastics on human health, either via direct exposure or trophic transfer in the ecosphere, remain unclear. This is especially true for nanoplastics, a conceivable end form of plastics through the forces of man and nature, whose biological implications have only begun to become apparent in recent years. In this study, we documented our discovery of endothelial leakiness induced by anionic polystyrene and PMMA nanoplastics, characterised by the techniques of confocal fluorescence imaging, transwell assays, signalling pathways with HUVCEs, rabbit and swine vessels, and mice, as well as virtual force microscopy of computer simulations. The endothelial leakiness showed dose and time dependence over cell exposure to both types of nanoplastics, mediated by conformational and structural changes to VE-cadherin junctions of endothelial cells engaged with the polymeric nanoparticles and independent of ROS production, autophagy, and apoptosis of the impacted cells. As the occurrence of endothelial leakiness implies that nanoplastics could permeate in and out of the vasculature that spreads the body, this discovery spells profound implications for understanding the safety and biological footprints of synthetic plastic materials crowding the environment.

## Methods
All animal experiments complied with ethical regulations for animal testing and research in accordance with the NIH's guidelines and were

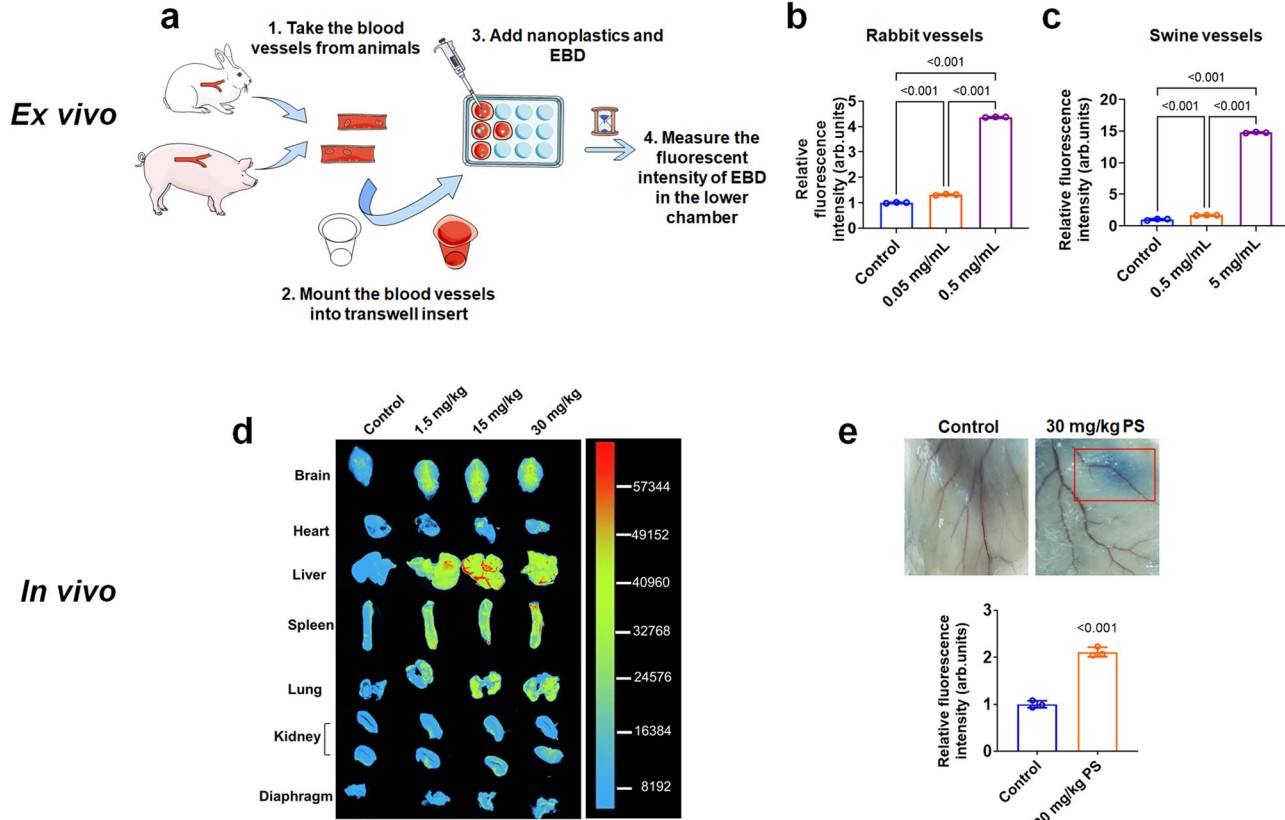

**Fig. 6 | Polystyrene nanoplastic-induced endothelial leakiness ex vivo in rabbit and swine vessels (a–c) and in vivo in mice (d, e). a** Schematic of the ex vivo construct. **b** Concentration-dependent increase in Evans blue dye (EBD) penetration in rabbit vessels. Quantifications of EBD indicated that the 0.05 and 0.5 mg/mL polystyrene (PS) nanoplastic groups were significantly different from untreated control. 0.5 mg/mL PS nanoplastic induced more leakiness than the 0.05 mg/mL PS nanoplastic group. Data are expressed as means ± SD. Biologically independent samples were used (*n* = 3). Statistical analysis was performed through one-way ANOVA followed by Tukey's multiple comparison tests. The derived *P* values comparing to control were inserted in the panel. **c** Concentration-dependent increase of EBD penetration in swine vessels. EBD penetration was more pronounced in the 5 mg/mL PS nanoplastic group than the 0.5 mg/mL PS nanoplastic group. Data are presented as means ± SD (*n* = 3 biologically independent samples), analysed via one-way ANOVA followed by Tukey's multiple comparison tests. The

derived *P* values comparing to control were inserted in the panel. **d** Male Swiss mice received intravenous injection of PS nanoplastic (1.5, 15, or 30 mg/kg)-containing 10 mM EBD solution. Control mice received once intravenous injection of 10 mM EBD. After 24 h, the mice were sacrificed to collect their organs for imaging. The fluorescence signal gradually increased from yellow to red with increased dose of the nanoplastic. The greater extent of EBD leakiness was, the stronger the fluorescence signal. **e** PS nanoplastic promoted leakiness of subcutaneous blood vessels in mice. PS nanoplastic (30 mg/kg) were injected into subcutaneous pockets on the back of the mice. EBD was injected via tail intravenous injection. Quantification of EBD showed more endothelial leakiness in the PS group compared with the untreated mice group. Data are expressed as means ± SD (*n* = 3 biologically independent animals), analysed via two-tailed Student's *t*-test. The derived *P* values comparing to control were inserted in the panel. Source data are provided as a Source Data file. (Some art elements in panel a are from smart.servier.com.).

approved by the Institutional Animal Care and Use Committee (IACUC) at Southwest University (reference (ID number) IACUC-20200525-01).

## Characterisations of nanoplastics

For TEM imaging of PS (30 nm carboxylated PS beads, catalog number L5155, Sigma-Aldrich, USA), NH₂-PS (30 nm aminated PS beads, catalog number: PSGF00030, Zhongke Detong Inc, China), and PMMA (50 nm carboxylated PMMA, catalog number: BKPMMA50, Beikenami Inc, China) nanoplastics, 5 μL of the samples was dropped on glow-discharged, formvar/carbon-coated copper grids (400 mesh, ProSci-Tech). After 1 min incubation, the grids were dried and washed with 10 μL of Milli-Q H₂O, then negatively stained with 5 μL of uranyl acetate (UA, 1 %). The samples were imaged with a FEI Tecnai F20 operated at 200 kV. The size distribution was analysed by ImageJ 1.53c software. The hydrodynamic diameters of the PS, NH₂-PS, and PMMA nanoplastics were determined by dynamic light scattering (DLS) with Zetasizer Nano-ZS (Malvern) at room temperature. The surface elemental compositions and chemical state of the nanoplastics were identified using X-ray photoelectron spectroscopy (XPS, Thermo escalab 250X, USA). The PS, NH₂-PS, and PMMA nanoplastics were

freeze-dried by a freeze dryer (Alpha2-42DPlus, Christ). Next the powder sample was glued to the sample stage with conductive adhesive, and then tested after vacuuming.

## Cell culture

Human umbilical vein endothelial cells (HUVECs, CRL-1730) were obtained from the American Type Culture Collection (ATCC, USA), and were cultured in complete endothelial cell medium (ECM, ScienCell, USA) containing 10% fetal bovine serum (FBS, Gibco, USA). Cells were incubated at 37 °C in a humidified atmosphere of 5% CO₂ in air. When the cells reached confluency, 0.25% trypsin was used to digest the cells for 4 min. Trypsin was then removed and serum-containing medium was introduced to the mixture to stop the digestion. Cells were pipetted down, centrifuged, and resuspended in a new medium.

## Flow cytometry analysis for nanoplastic uptake

Approximately 1 × 10⁶ cells/well were seeded in 6-well plates overnight followed by the treatment of PS nanoplastic or NH₂-PS nanoplastic at 0.05 or 0.5 mg/mL for 1, 3, or 6 h. The cells were then washed with PBS and analysed using a BD FACS Melody™ flow cytometry. A total of

$1 \times 10^4$ events was acquired for each sample from three independent experiments. The data were analysed by FlowJo_V10.

### ROS level measurement

Approximately $1 \times 10^6$ cells/well were grown on 6-well plates and exposed to the PS nanoplastic (0.05, 0.1, 0.25 and 0.5 mg/mL) for 1, 3 or 6 h. Cells in the control group received serum-free medium treatment for 1, 3, or 6 h and were then incubated with DCFH-DA (Sigma-Aldrich, USA) indicator in the dark at 37 °C for 30 min. Cells were washed twice with phosphate buffered saline (PBS). The cellular fluorescence was analysed by a BD FACS Melody™ flow cytometer. Experiments were repeated 3 times independently. The data were analysed by FlowJo_V10.

### Cell viability assay

HUVECs ($5 \times 10^4$ cells/well) were seeded in a black, clear bottom 96-well plate (Corning Costar, USA) and cultured at 37 °C for 48 h. The old media were replaced with fresh ECM containing $1 \times 10^{-6}$ M propidium iodide (PI, Sigma-Aldrich, USA). After 30 min incubation, cells were exposed to different concentrations of PS nanoplastic (0, 0.05, 0.1, 0.25 and 0.5 mg/mL). Subsequently, cell morphology and number of dead cells were measured using Operetta (20× PlanApo objective, numerical aperture NA = 0.7, PerkinElmer) with a live cell chamber (37 °C, 5% CO$_2$) over 22 h. The cell mortality was calculated by PI-positive cells to total cell counts, which were derived from the bright-field mapping function of Harmony High-Content Imaging and Analysis software (PerkinElmer). HUVECs ($5 \times 10^3$ cells/well) were seeded in a 96-well plate overnight before treatment with PS, NH$_2$-PS, and PMMA nanoplastics (0.05 and 0.5 mg/mL) for 1 h, 3 h, and 6 h. Negative control was exposed to fresh medium for 1, 3, or 6 h. Positive control was exposed to 200 μM H$_2$O$_2$ for 1, 3, or 6 h. Subsequently, A volume of 100 μL medium and 10 μL CCK8 reagent were added to each well and incubated at 37 °C for 2 h. The absorbance of each culture well was measured with a microplate reader (BioTek, USA) at the wavelength of 450 nm.

### Western blotting assay

HUVECs at a concentration of $1 \times 10^6$ cells/well were seeded in 6-well plates (Dingjin, China). For autophagy signaling pathway, cells were treated with PS nanoplastic (0.05 or 0.5 mg/mL) for different times (1, 3, or 6 h), negative control was exposed to fresh medium for 1, 3, or 6 h. For PS nanoplastic (0.05 or 0.5 mg/mL) treatment for 6 h, negative control was exposed to fresh medium for 6 h. For apoptosis signaling pathway assay, cells were treated with PS nanoplastic (0.05 or 0.5 mg/mL) for 6 h, and negative control was exposed to fresh medium for 6 h. For VE-cadherin signaling pathway assay, cells were treated with Src kinase inhibitor PP1 (10 μM) for 1 h prior to PS nanoplastic (0.05 or 0.5 mg/mL) treatment for 1, 3, or 6 h. Groups without PP1 were exposed to fresh medium with only PS nanoplastic (0.05 or 0.5 mg/mL) treatment for 1, 3, or 6 h. Negative control was exposed to fresh medium. For VE-cadherin signaling pathway assay, cells were treated with PMMA nanoplastic (0.05 or 0.5 mg/mL) for 1, 3, or 6 h. After washing 3 times with PBS, the cells were lysed with lysis buffer on ice for 10 min. For VE-cadherin signaling pathway assay in ex vivo models, rabbit and swine vessels were homogenised at 4 °C and centrifuged at 10,000 g, 4 °C for 20 min, and the supernatants were collected. The protein concentrations of cell lysates were determined using Carmassi Bradford method. Cellular proteins from each group were separated by standard 8% or 12.5% or 15% sodium dodecylsulphate polyacrylamide gel electrophoresis (SDS-PAGE) and then blotted onto nitrocellulose membranes (PALL, USA). The membranes were blocked with 5% skim milk at room temperature for 2 h, and then incubated with a corresponding primary antibody at room temperature for 3 h. The membranes were washed with Tween 20 (Biofroxx, Germany)/Tris-buffered saline (TBST) and incubated with biotinylated goat anti-rabbit IgG (H + L) or biotinylated goat anti-mouse IgG (H + L) antibody (1:2000 dilution, Sangon, China) for 1 h and with HRP-labeled streptavidin antibody (1:5000 dilution, Sangon, China) for 1 h at room temperature. The membranes were visualised in the electrogenerated chemiluminescence (ECL) system. Representative images were chosen from at least three independent experiments. The images of protein bands were analysed in a semi-quantitative manner through ImageJ 1.53c software. The uncropped scans of the blots were provided in the Source Data file. The primary and secondary antibodies were listed in Supplementary Table 2.

### RNA extraction and real-time quantitative PCR (RT-qPCR)

HUVECs at a concentration of $1 \times 10^6$ cells/well were seeded in 6-well plates (Dingjin, China). Cells were treated with PS nanoplastic (0.05 or 0.5 mg/mL) for 6 h. Then cells were lysed and collected for RNA extraction with a total RNA purification kit (BioTeke, China). The purified RNA was reverse transcribed into cDNA with the transcriptor first strand cDNA synthesis kit (Yeasen, China). Next, cDNA was analysed by RT-qPCR analysis using 2× SYBR Green qPCR Master Mix (Bimake, USA) in a real-time PCR machine (Roche, Switzerland). Data from three independent experiments were analysed with β-actin as the internal standard. The primers used for the amplification were listed in Supplementary Table 3.

### DAPI staining

HUVECs were seeded in confocal dishes at a density of $8 \times 10^4$ cells/well and cultivated for 24 h. Cells were treated with PS nanoplastic (0.05 or 0.5 mg/mL) for 6 h. Then, cells were stained with DAPI (200 μL) for 10 min. After washing with PBS three times, morphological features of the cells were captured using a confocal fluorescence microscope (Nikon, N-SIM E).

### Confocal fluorescence microscopy

HUVECs ($5 \times 10^5$ cells/well) were seeded on 8 well chamber slides and cultured 5–6 days to form an intact monolayer. Then PS nanoplastic, NH$_2$-PS, and PMMA at 0.05 and 0.5 mg/mL in fresh cell media were added into the wells and incubated with cells for 1, 3, or 6 h. After that, the cells were washed twice carefully with Hanks' Balanced Salt Solution (HBSS, Sigma Aldrich, USA) and further fixed by 4% paraformaldehyde (PFA, Sigma Aldrich, USA) for 15 min at room temperature. After washing with PBS, the cells were blocked using 0.1% saponin and 5% horse serum in PBS for 1 h at room temperature. Subsequently, immunostaining was utilized to elicit the expression of VE-cadherins and actin filaments. Rabbit polyclonal anti-VE-cadherin antibody at 1:400 in 5% horse serum/PBS was added into the chamber wells and incubated overnight at 4 °C. After washing with PBS, cells were incubated with secondary antibody solution (Donkey anti-rabbit Alexa Fluor 594 at 1:500 in PBS containing 0.1% phalloidin-iFluor 488) for 2 h at room temperature. Then Hoechst 33342 (Thermo Fisher Scientific, USA) at 1:2,000 in PBS was added to cells and incubated for 5 min. The chamber slides were then imaged using a confocal fluorescence microscope (SP8 LIGHTNING, Leica Microsystems) with 63× oil objective. Semi-quantitative image analysis for gap area and actin intensity was performed using ImageJ 1.53c software.

### In vitro transwell assay

HUVECs were seeded in transwell inserts (polycarbonate membrane, 0.4 μm pore diameter; Corning Costar, USA) with approximately $8 \times 10^4$ cells and grown for 4 days to reach ~100% confluency to form an intact monolayer. Then the cells were treated with PS (0.05, 0.1, 0.25 and 0.5 mg/mL), NH$_2$-PS (0.05 and 0.5 mg/mL) or PMMA (0.05 and 0.5 mg/mL) nanoplastic for 1, 3 or 6 h. For pharmaceutical inhibition assay, cells were treated with corresponding inhibitors, endocytosis inhibitor 5 mM methyl-β-cyclodextrin (MβCD, Solarbio, China) or 10 μM monodansylcadaverine (MDC, Sigma Aldrich, USA), 10 μM

ROCK inhibitor Y-27632 (Yuanye, China), 10 μM Src kinase inhibitor PP1 (Yuanye, China) or 5 mM ROS inhibitor N-Acetyl-L-cysteine (NAC, Sigma Aldrich, USA) for 1 h prior to PS nanoplastic (0.05 or 0.5 mg/mL) treatment for 1 h. Groups without the inhibitors were exposed to fresh medium with only PS nanoplastic for 1 h. Negative control was exposed to fresh medium for 1 h. After the treatment, the cells were washed twice carefully with PBS. FITC-dextran (average molecular weight 40,000, Sigma-Aldrich, USA) of 100 μL with a final concentration of 1 mg/mL was added to each well and the media from the lower compartment were collected after 30 min. Fluorescence intensities were measured at excitation/emission of 495 nm/519 nm with a microplate reader. Readout from the negative control group was used to normalise the other groups' readouts.

### Nanoplastic and HUVECs interaction assay

HUVECs were seeded in a 96-well plate (black/clear bottom, Corning Costar, USA) with approximately $5 \times 10^4$ cells/well and incubated at 37 °C for 48 h. Then cells were treated with PS nanoplastic (0.05, 0.1, 0.25 and 0.5 mg/mL) for different times (1, 3 and 6 h). Cells in the control group received serum-free medium treatment for different times (1, 3, and 6 h). Intracellular fluorescence was measured using Operetta (PerkinElmer, 20× PlanApo microscope objective, numerical aperture NA = 0.7).

### Nanoplastic binding to adherens junctional homophilic VE-cadherins

Confluent monolayers of HUVECs were treated with PS nanoplastic (0.05 and 0.5 mg/mL) for different times (1, 3, and 6 h). Then the proteins were extracted with lysis buffer. PS nanoplastic with their associated proteins were pulled down by centrifugation (10,000 $g$, 20 min, 4 °C). PS nanoplastic were washed 3 times with RIPA buffer, and the associated proteins were denatured from the PS nanoplastic at 97 °C for 10 min in loading buffer. For post lysis pull-down experiment, HUVECs were lysed and were added with PS nanoplastic (0.05 and 0.5 mg/mL) or Protein A/G magnetic beads (Bimake, USA). Then the mixtures were gently shaken for 1 h at 4 °C. Afterward, PS nanoplastic or Protein A/G magnetic beads with their associated proteins were pulled down by centrifugation (10,000 $g$, 20 min, 4 °C) and washed 3 times with RIPA buffer. The proteins were separated by standard 8% SDS-PAGE and then transferred onto polyvinylidene fluoride (PVDF) membranes. Then membranes were probed with antibodies against VE-cadherin (1:1000; Abcam, USA) and α-tubulin (1:2000; Proteintech, China).

### Discrete molecular dynamics

All-atom discrete molecular dynamics (DMD) has been widely used for biomolecular studies including protein folding, peptide aggregation[38–40], protein structure and dynamics[41,42], and nanoparticle-protein interactions[21,43,44]. DMD is a special class of molecular dynamics (MD) where the force field is remodeled as discrete step functions from conventional MD simulations[45]. Specifically, the inter-atomic potential of the DMD simulations in this study consisted of the bonded (i.e., bonds, bond angle, and dihedral angle) and non-bonded (i.e., van der Waals, electrostatic, solvation, and hydrogen bonds) terms. In non-bonded terms, CHARMM forcefield[46] and Debye-Hückel approximation were applied to van der Waals and the electrostatic terms, respectively. Solvation in non-bonded parameters was employed by the EEF1 implicit solvent model developed by Lazaridis and Karplus[47], and hydrogen bond was modeled with the reaction-like algorithm[48].

### DMD simulations for polystyrene and PMMA nanoplastics binding with cadherin dimers

Cadherin is a calcium-dependent protein, which is one of the main proteins for cell-to-cell adhesion. It has been known that the *trans* dimer is formed by the cadherin proteins coming from two opposing cells involved in cell-to-cell adhesion. We focused on the *trans* interaction that was mainly formed by the domain-swapped region in the first extracellular domain (EC1) domains. We used the EC1 cadherin dimer from the cryo-EM model of EC12 cadherin dimer (PDB ID: 3PPE[49]). To ensure the structural stability of the native EC1 dimer, the domain-swapped region including the N-terminal (i.e., residue 1–5) was constrained by applying the Gō-potential respectively (Fig. 5a). Here, the Gō constraints were assigned only between the $C_\beta$ atoms of contacting residues with a weak contact energy of 0.3 kcal/mol (~0.5 $k_B$T). The $Ca^{2+}$ coordination of the loops (i.e., residues Glu11, Asp62, Glu64, Asp96, and Asp99) were modeled by pairwise bonded constraints among the coordinated oxygen atoms. Next, we constructed a PS and PMMA nanoplastic with carboxyl surface groups for each. A PS chain composed of repetitive 20 styrene monomers was prepared and relaxed and equilibrated for 50 ns DMD simulations. Once the pure PS chain formed a compact particle, carboxyl groups were added to the PS chain followed by 50 ns DMD simulations. The PMMA nanoplastic was modeled following the method for the PS nanoplastic. Constructed long PS and PMMA chains with carboxyl groups maintained as a compact structure with the carboxyl groups exposed (Fig. 5b and Supplementary Fig. 16a). Subsequently, the nanoplastic randomly distributed near the EC1 cadherin dimer at least 12 Å away in a 120 nm³ cubic box to perform the binding simulation. During the simulations, the backbone of the EC1 dimer was constrained and their sidechains freely interacted with the PS and PMMA nanoplastics. For running DMD simulations, 50 fs/step of the unit simulation time and 1 kcal/mol of corresponding energy were employed and a temperature of 300 K was maintained with Anderson's thermostat. We carried out 30 independent DMD simulations for 50 ns and an accumulative 1.5 μs total simulations. After the binding simulations, the binding frequencies of the PS and PMMA nanoplastics with the EC1 cadherin dimer were calculated from the last 20 ns of the simulations. A cutoff distance of 0.65 nm was assigned to get an atomistic contact between the EC1 cadherin dimer and the PS and PMMA nanoplastics to calculate their binding frequency.

### Steered discrete molecular dynamics (sDMD) simulations

Next, constant force-pulling in silico experiments on cadherin-PS and cadherin-PMMA nanoplastic complexes were performed to identify the EC1-EC1 cadherin dimer stability via steered molecular dynamics (sDMD) simulations. The sDMD technique generally mimics optical tweezers and atomic force microscopy (AFM) to characterise biomolecules as a function of either a constant velocity or a constant force. The sDMD simulation approach has successfully characterised protein-ligand binding, protein-nanoparticle complexes[21,50], and protein unfolding[51].

Prior to sDMD simulations, we distributed chloride ions near the cadherin-PS nanoplastic complex to neutralise the charges and randomise the initial velocity of all atoms of the system. Subsequently, we immobilised the backbone of one of the EC1 dimers (excluding the domain-swapped fragment) and the rest of the domain was flexible (Fig. 5e). The flexible domain from the EC1 cadherin dimer was pulled by constant forces along the EC1 to EC2 direction mimicking the *trans*-interaction of the cadherin dimer during the sDMD simulations. Details of running sDMD simulations are the same as the binding DMD simulations. We applied 10 pN as an interval force within the range of 0 to 20 pN. For sufficient sampling, 30 cases of independent sDMD simulations each for 100 ns were performed. We used pyMol (Schrödinger) to analyse the dissociation trajectories of cadherin dimer with and without nanoplastics.

### Measurement of EC1 cadherin dimer angle

The various species of cadherin have two different intermediate and domain-swapped dimers known as x-dimer and s-dimer during the

dimerisation. It has been identified that the dimer angles of x-dimer and s-dimer from the crystal structure of the cadherins are ~0 and 90 degrees with respect to each other[21]. We calculated the EC1 dimer angle without the applied force for the first 30 ns of the sDMD simulations before the dimer started to dissociate. The dot products of two vectors between the 88th and 98th residues for s-dimer and 92th and 99th for x-dimer from both the flexible and immobilized domains of the EC1 cadherin dimer were considered. The dimer angle was calculated using MATLAB_R2017a, and corresponding data analysis was performed using Python, visual molecular dynamics (VMD), and Grace.

### Ex vivo transwell assay
A vascular leakiness assay was performed using rabbit and swine vessels as transwell inserts. Vessels of three 8-month-old male Large White Swine were obtained from a local slaughterhouse in Chongqing, China. Vessels of three 4-month-old male New Zealand Rabbits were obtained from Ensiweier Biotechnology Co, Ltd. (Chongqing, China). Vessels of the coronary artery were cut transversely into individual pieces and placed in a commercial transwell chamber after removal of its original membranes. PS nanoplastic of 0.5 and 5 mg/mL were added to the custom-made rabbit or swine vessel transwell device and then incubated for 3 or 6 h at 37 °C, respectively. After the exposure, the PS nanoplastic-containing solution was discarded and then 100 mM EBD (Macklin, China) was added to each well, incubated for another 1 h. Finally, the fluorescence signal from the lower compartment of the transwell was quantified at 624 nm with a microplate reader. Readouts from the negative control group were used for normalisation.

### In vivo leakiness assay
Twelve 10-weeks-old male Swiss mice were obtained from Ensiweier Biotechnology Co, Ltd. (Chongqing, China). Male Swiss mice were supplied with free access of food and water and were kept in a standard temperature and humid environment with a light/dark cycle of 12 h. The mice received once intravenous injection of PS nanoplastic (1.5, 15, or 30 mg/kg)-containing 10 mM EBD solution. The control mice received once intravenous injection of 10 mM EBD solution. After 24 h, the mice were sacrificed to collect the organs for imaging using NEWTON 7.0 Imaging System.

### In vivo subcutaneous vascular leakiness assay
Six 10-weeks-old male Swiss mice were obtained from Ensiweier Biotechnology Co, Ltd. (Chongqing, China). All in vivo mice experiments were carried out in accordance with the NIH's guidelines for the care and use of laboratory animals and were approved by the Southwest University Animal Care and Use Committee. Male Swiss mice were supplied with free access to food and water and were kept in a standard temperature and humid environment with a light/dark cycle of 12 h. Male Swiss mice received once intravenous injection of 10 mM EBD solution. Then mice were anesthetised with isoflurane and subcutaneous injection with either PBS or 30 mg/kg PS nanoplastic. The mice were sacrificed by cervical dislocation 15 min after they had received subcutaneous injections. Then the skin regions were excised to extract extravasated dye by deionized formamide. Finally, 100 μL of the dye-containing supernatant from each sample was transferred into a separate well of 96-well plate. The fluorescence signals from the samples were quantified at 624 nm with a microplate reader.

### Statistics and reproducibility
The in vitro assays, including cell viability, ROS production, in vitro and ex vivo transwell assays, RNA extraction, Western blot, confocal microscopy imaging, were derived from at least three biologically samples ($n = 3$). The extent of endothelial leakiness was expressed by gap area and distribution, which were derived from the images using trainable Weka segmentation plugin in ImageJ 1.53c software.

The data graphing and statistical analysis were performed with GraphPad Prism version 9.3.1 (GraphPad Software, La Jolla) and Origin 7.5. All the data were expressed as mean ± standard deviations (SD), analysed via one-way or two-way ANOVA, and followed by Tukey's multiple-comparisons test as indicated in the respective figure captions. $P < 0.05$ was considered statistically significant.

### Reporting summary
Further information on research design is available in the Nature Research Reporting Summary linked to this article.

## Data availability
The source data underlying the respective main text (Figs. 1–4, 6) and Supplementary Figures (Supplementary Figs. 1, 2, 4, 5, 9–13, 18, and 19) are provided as Source Data File. Source data are provided with this paper. The source data for Fig. 5 and Supplementary Figures 14–17 were deposited in the webserver https://dlab.clemson.edu/research/NanoPlasticEL/. The description data for s-dimer and x-dimer were deposited in Protein Data Bank (PDB) under the accession codes 3PPE and 4ZT1. Source data are provided with this paper.

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

## Acknowledgements

This work was supported by the National Key R&D Program of China (2021YFA1200900 C.C. and P.C.K.), Department of Science and Technology of Guangdong Province (2019GD0101 C.C.), National Natural Science Foundation of China (21976145 Y.S., 21974110 Y.S., and 82104087 Y.L.), National Science Foundation (CAREER CBET-1553945 F.D.) and National Institutes of Health (MIRA R35GM119691 F.D.).

## Author contributions

P.C.K. conceived the project. P.C.K., D.T.L., Y.S., F.D., C.C., and S.L. contributed to the project design. Y.L., W.W., M.L., and P.C.K. wrote the manuscript. W.W. and Y.S. performed in vitro and ex vivo transwell, signaling pathways, ROS, autophagy, apoptosis, and in vivo assays. Y.L. and N.A. performed TEM, DLS, zeta potential, confocal fluorescence microscopy, transwell, as well as gap area analysis. M.L. and F.D. performed DMD and sDMD simulations and analysis. Y.L. performed overall experimental data analysis and figure representation. All authors discussed and agreed on the presentation of the manuscript.

## Competing interests

The authors declare no competing interests.
