## [Peer Review File · Nature Communications]

REVIEWER COMMENTS

Reviewer #1 (Remarks to the Author):

The manuscript written by Yang Song, Pu Chun Ke and coauthors presents experimental results on the effects of carboxylated polystyrene beads, smaller than 50 nm, on the vasculature permeability, induced by the disruption of the vascular endothelial cadherin junctions and independent of cytotoxicity. Authors study the biophysical molecular mechanism of VE-cadherin dimer rupture induced by carboxylated polystyrene nanoplastic using molecular dynamics simulations, and they verify the endothelial leakiness by ex vivo and in vivo experiments. The effects of nanoplastic exposure on tissues are extremely important and difficult to investigate. This experimental work can be significant in this field, but it requires revision before publication, as detailed below.

1. As some of the authors reported in their previous works, anionic inorganic nanoparticles (NPs) can migrate to and disrupt the adherens junctions between endothelial cells to induce NanoEL. The nanoplastic selected for the in vitro studies are 30 nm carboxylated PS beads by Sigma-Aldrich.

Why did you perform experiments only on carboxylated beads? The interaction with cell membranes could be affected by the negative charge of nanoparticles, inhibiting the internalization and/or the toxicity. Also the presence of charges could modulate the interaction of PS beads with VE cadherin.

Please discuss this point and add experimental evidence on different polystyrene nanoparticles.

2. "The morphology and hydrodynamic diameter of the polystyrene (PS) nanoplastic were characterised by transmission electron microscopy (TEM) and dynamic light scattering (DLS). As displayed in Figure 1A&B, the PS beads were mono-dispersive with an average size of 21.2 ± 3.5 nm quantified by TEM. In comparison, the hydrodynamic diameter of the PS nanoplastic was 43.9 ± 0.4 nm (Figure S1)."

Figure S1 shows the volume distribution of the hydrodynamic diameter of PS nanoparticles. Authors report the mean size and, I suppose, the standard deviation obtained performing several measurements. This is not the interesting parameter to define the monodispersity of the system. Rather the polydispersity index has to be reported. Looking at figure S1, I think that the polydispersity is large. Can you comment on the polydispersity of PS nanoparticles?

Also, can you discuss the discrepancy between TEM and DLS results?

3. “Next, PS nanoplastic with carboxyl surface groups were constructed to investigate their competence with NanoEL. Due to the high computational cost related to the large size of PS, we modeled PS nanoplastic composed of 20 repetitive PS chains and the diameter of the formed nanoplastic was equivalent to ~ 15 Å for the 50 ns equilibrium DMD simulation.”

The behaviour of PS chains of different lengths in interaction with model cells is of great interest. PS chains can interact with cells, penetrating plasma membranes. Authors could complete the biography with recent papers reporting computational investigations on PS chains of similar length (G. Rossi et al., Polystyrene Nanoparticles Perturb Lipid Membranes, *J. Phys. Chem. Lett.* 2014, 5, 241–246; D. Bochicchio et al., Polystyrene perturbs the structure, dynamics, and mechanical properties of DPPC membranes: An experimental and computational study, *J. Colloid Interface Sci.* 605 (2022) 110–119).

A critical point is the direct comparison between in silico results on 20-mers PS and in vitro experimental results on PS commercial beads (30 nm).

Authors have to discuss this severe limitation to the study.

4. “These regions were located away from the interfacial region of the dimer and were rich in positively charged residues, as indicated by the surface representation of the dimer color-coded according to the nanoplastic-binding frequencies. The binding frequency data suggested that the PS nanoplastic preferred to bind near the turn region of the dimer. Therefore, the electrostatic interactions between led to association of the two entities.”

In silico results reveal that carboxylated PS chains interact with the dimer of VE cadherin through electrostatic interaction between the carboxyl groups on the surface of the PS nanoplastic and the positively charged residues in the turn region of the dimer, as reported in Figure 4. Authors must exclude that the interaction with VE cadherin is governed solely by the charge of the polymer, by carrying out tests both on negatively charged polymer chains other than plastics and on non-derivatized PS chains, to elucidate the intrinsic propensity of PS, if any, to interact directly with the dimer of VE cadherin leading to dimer dissociation.

Reviewer #2 (Remarks to the Author):

The manuscript presents research of the potential for nanoplastics to induce endothelial leakiness. The hypothesis proposed has important implications for humans and other organisms with a vascular system

and major significance given the pervasiveness of nanoplastics in the environment. The study is detailed, comprehensive and well documented. However, it completely relies on just one type (and size) of nanoplastic, which is not necessarily the most common in the environment and, as a purpose made plastic, may not be representative of other environmental nanoplastics such as those produced by fragmentation and ageing of bulk plastics. This should at the very least be discussed in the paper and the claims made should be moderated accordingly.

The manuscript is mostly well written but in places requires attention, as follows:

Line 83: replace “vines” with “veins”

Line 92: replace “mono-dispersive” with “monodisperse”

Lines 99-108: the discussion about ROS generation and cell death is written in a rather confusing way and requires revision.

Line 110: replace “microtuble” with “microtubule”

Lines 308-309: sentence starting “As the concentration...” needs revision

Line 351: replace “reveal” with “become apparent”

Lines 358-362: wording and claim made in this sentence needs revision (see also earlier discussion).

Lines 394-396: Sentence starting “Subsequently...” needs revision

Line 409: Sentence starting “Then cells...” needs revision

Reviewer #3 (Remarks to the Author):

NCOMMS-21-48035

Nanoplastic Exposure Induces Endothelial Leakiness

The manuscript from Wei et al. examines whether acute exposure to polystyrene nanoplastics (nPS) can disrupt the endothelial barrier, along with an assessment of the potential mechanisms of action.

‘Microplastics and nanoplastics exposure and effects’ is currently a very important research topic for researchers, policy-makers, regulators, industry, citizens, and society in general, since evidence indicate

that humans are inevitably exposed, and that plastic particles have the potential for biodistribution and bioaccumulation inside the body to produce negative health consequences.

Most of this evidence has been obtained using ecotoxicity-related experimental models, therefore data from more relevant human experimental models such as the in vitro vascular endothelial system used here by the authors are at present very valuable. The hypothesis of nPS being able to alter the functionality of the vascular endothelium is interesting, novel, attractive, and supported by the fact that other particulate materials (nanomaterials) are known to cause this same effect. It is also relevant that nanoplastics can translocate through the gastrointestinal barrier to reach blood vessels. Thus, the vascular endothelium is an interesting secondary target. It is also of note that the health implications of nanoplastic causing endothelial leakiness are huge.

Despite of the very high interest of the approach, the experimental data presented here is not convincing enough to recommend publication. Below are the main points that justify my decision:

-The doses of nPS are unusually high. Most of them fall in what is considered non-biologically relevant. For some of the endpoints, the authors use 1 milligram per milliliter. Even though there are not environmental/human estimates for nanoplastic exposure, it is very unlikely that these concentrations can reach the blood system. One could justify that using non-biological concentrations is still useful for risk assessment purposes, and that would at some point be acceptable, but the problem with nanoplastics (and nanomaterials in general) is that technical issues start to appear when using concentrations above 100 micrograms per milliliter: the material aggregates massively and deposit on the surface of the plate, covering the cells and preventing from normal cell growth and behavior. Any effect found under these conditions is irrelevant.

-Authors culture HUVECs in RPMI media + 10% FBS, which is not recommended. RPMI is low in cations. This may affect cell interactions that are cation-dependent, such as adherens junctions. Hence, endothelial leakiness could have been influenced by the culture media conditions.

-Authors poorly assess nPS internalization, localization in the cytosol, or the membrane fraction. The nPS used is fluorescent, hence a better assessment using confocal microscopy and flow cytometry is expected.

Also regarding nPS internalization, Figure S5 is presented in a way where time-dependent internalization cannot be evaluated.

-According to Figure 1- brightfield image, the concentration of 1 milligram per milliliter dose is causing extensive cell death (maybe due to technical issues explained in the previous point?), moreover not matching with Figure 1E.

-nPS characterization should be evaluated in cell growth media (and not only milliQ water) to have a better estimate of the sizes and zeta potentials of the nPS in a media that better mimic the treatment conditions. Please indicate how many TEM images have been analyzed to determine the particle size.

-ROS data lacks positive control and time-matched controls. ROS is quite high at concentrations where no toxicity is found. Is this expected? The variability of ROS values seems low for this technique. A complementary technique to assess ROS is necessary.

-Why have authors explored the mechanisms of nPS-induced cell death (autophagy and apoptosis) at time-points where cell death is not evident? Why they have not included the 22h where the maximum mortality rate is found?

-Autophagy data lacks positive control and negative time-matched controls.

-The manuscript contains some contradictory statements when explaining whether authors use serum-free media for treatments or not, and the same for the concurrent controls. This needs to be clarified. Importantly, serum deprivation induces autophagy. Hence, reasonable doubts arise about nPS being the cause of autophagy induction.

-Authors conclude that nPS is not toxic under 0.5 mg/mL and 3 h of exposure. Based on what? Only cell mortality data? ROS and autophagy markers are positive at time-points and concentrations where they claim there is no toxicity. And we do not know about apoptosis because only 6h is shown.

-Authors assess endothelial leakiness at time-points and concentrations where ROS and apoptosis are evident.

-It is not clear how authors assess that they have generated an intact monolayer in the transwell before applying the nPS treatment.

-Negative control in Figure S6 should be included in the main Figure. Include also the rest of time-matched negative controls. Holes in the monolayer of the shown control are evident. Is this expected? The same for 1h of treatment in Figure 2C, although there are no arrows pointing to them. The quality of the negative control and the 1h of treatment pictures is lower than the rest. Overall, no solid data/images support the leakiness effect.

-Why the authors didn't use a positive control for endothelial leakiness such as EDTA?

-To assess the role of ROS and endocytosis, authors selected 1h of nPS treatment to assess endothelial leakiness in the transwell with and without the use of inhibitors or NAC (Figure 3). Why? According to Figure 2, no FITC-dextran was detected in the basolateral compartment at this condition.

-Authors modeled a 1.5 nm nPS particle due to the computational cost. Is then the result comparable to nPs particles of 30 nm? This needs to be discussed/clarified.

-Indicate the equivalences of the in vivo selected dose.

-Dose-dependency is not found in most of the organs after the in vivo EDB+nPS assay. This needs to be discussed.

-A complementary in vivo analysis of direct vascular leakiness should be included to reinforce the findings i.e. inject nPS or vehicle into subcutaneous pockets followed by EBD injection.

-Scheme in Figure S2A is not very helpful.

-Results and discussion section contain too many technical details that should be moved to methods or figure legends. Also, this section will benefit from including more statements regarding interpretation and discussion of the data.

Reviewer #4 (Remarks to the Author):

This study investigated the potential toxicity of polystyrene nanoplastic (PS nanoplastic) to vascular endothelium. The results showed that vasculature permeability could be a new mechanism for the transport of nanoplastics. A series of experiments were conducted to investigate the underlying mechanisms, which revealed that exposure to PS nanoplastic induced phosphorylation of VE-cadherin, and further induced endothelial leakiness. The authors performed several validating experiments to

make their conclusion reliable. However, there are a number of concerns. Revisions are required before consideration of publication. Specific comments are provided below.

1. Carboxylated PS beads were chosen in this study. Is this the main form of polystyrene nanoplastic in nature?

2. Could all the nanoparticles (around 30 nm) contribute to this phenotype. In other words, Did the phenotype in this study is PS nanoplastic specifically induced?

3. p62 is a receptor for cargo destined to be degraded by autophagy.

Line 114-116, why both expression level of LC3-II and p62 were increased at 3 and 6 h? These results should be discussed.

4. Line 121-122: "We therefore concluded that PS nanoplastic under 0.5 mg/mL and within 3 h exposure would not cause significant toxicity" . There several markers were changed in 3 hours, such as ROS, autophagy, etc. So more markers should be detected to support this conclusion.

5. More experiments should be performed to confirm that the PS nanoplastic really bound the VE-cadherin

6. The phosphorylation of VE-cadherin should be also detected in the Ex vivo and In vivo model.

7. There are still studies conducted on rodents or on cell cultures that would be relevant to mention on the introduction and to consider as comparison on the discussion section:

<https://www.tandfonline.com/doi/abs/10.1080/17435390.2021.1930228?journalCode=inan20>

Reviewer #1 (Remarks to the Author):

The manuscript written by Yang Song, Pu Chun Ke and coauthors presents experimental results on the effects of carboxylated polystyrene beads, smaller than 50 nm, on the vasculature permeability, induced by the disruption of the vascular endothelial cadherin junctions and independent of cytotoxicity. Authors study the biophysical molecular mechanism of VE-cadherin dimer rupture induced by carboxylated polystyrene nanoplastic using molecular dynamics simulations, and they verify the endothelial leakiness by ex vivo and in vivo experiments. The effects of nanoplastic exposure on tissues are extremely important and difficult to investigate. This experimental work can be significant in this field, but it requires revision before publication, as detailed below.

We thank the reviewer for strong endorsement of our manuscript for its significance.

1. As some of the authors reported in their previous works, anionic inorganic nanoparticles (NPs) can migrate to and disrupt the adherens junctions between endothelial cells to induce NanoEL. The nanoplastic selected for the in vitro studies are 30 nm carboxylated PS beads by Sigma-Aldrich.

Why did you perform experiments only on carboxylated beads? The interaction with cell membranes could be affected by the negative charge of nanoparticles, inhibiting the internalization and/or the toxicity. Also the presence of charges could modulate the interaction of PS beads with VE cadherin.

Please discuss this point and add experimental evidence on different polystyrene nanoparticles.

Thank you for this comment. The negative charged carboxylated-PS nanoplastic were repelled by the net negative cell membranes to discourage endocytosis but favoured their entry into the paracellular space to induce NanoEL.

As the most used model nanoplastic in literature (~98% of the publications we have surveyed), carboxylated PS nanoplastic possess good stability and water suspensibility, which guaranteed a higher repeatability of the experiments. Per the reviewer's suggestion, we have added measurements for two other types of nanoplastics, aminated PS (NH₂-PS) and poly(methyl methacrylate) (PMMA). Our transwell and confocal data (Figs. 2&3) have demonstrated that NanoEL could be induced by both negatively charged nanoplastics of carboxylated PS and

PMMA, but not the positively charged nanoplastic (NH₂-PS). Specifically, we did not observe NanoEL with aminated-PS nanoplastic on HUVEC cells at different concentrations and time points, indicating that charge could modulate the interaction of PS nanoplastic with VE-cadherin, consistent with the findings of NanoEL with inorganic nanoparticles.¹

In addition, we have provided experimental data on the internalisation and toxicity of both types of PS and PMMA nanoplastics in Figs. 1, 3 and S9d. The negatively charged PS induced less internalisation compared to positively charged NH₂-PS via flow cytometry in Figs. 1f&S9d. NH₂-PS was more taken up by cells rather than inducing NanoEL.

2. “The morphology and hydrodynamic diameter of the polystyrene (PS) nanoplastic were characterised by transmission electron microscopy (TEM) and dynamic light scattering (DLS). As displayed in Figure 1A&B, the PS beads were mono-dispersive with an average size of 21.2 ± 3.5 nm quantified by TEM. In comparison, the hydrodynamic diameter of the PS nanoplastic was 43.9 ± 0.4 nm (Figure S1).”

Figure S1 shows the volume distribution of the hydrodynamic diameter of PS nanoparticles. Authors report the mean size and, I suppose, the standard deviation obtained performing several measurements. This is not the interesting parameter to define the monodispersity of the system. Rather the polydispersity index has to be reported. Looking at figure S1, I think that the polydispersity is large. Can you comment on the polydispersity of PS nanoparticles? Also, can you discuss the discrepancy between TEM and DLS results?

Thanks for this question. The PDI values for the three types of nanoplastic measured by DLS in H₂O and in endothelial cell media have been summarised in Table S1. TEM measurements revealed the actual morphology and size of the nanoparticles, while DLS reflected the hydrodynamic diameters of the nanoparticles in diffusion. The hydrated layer on the nanoparticle surface impacted their dynamic light scattering during the test, leading to larger diameter readings than the TEM measurements which were performed in dehydrated state. We have added a note on this in the revised manuscript, on p4.

3. “Next, PS nanoplastic with carboxyl surface groups were constructed to investigate their competence with NanoEL. Due to the high computational cost related to the large size of PS, we modeled PS nanoplastic composed of 20 repetitive PS chains and the diameter of the formed nanoplastic was equivalent to ~ 15 Å for the 50 ns equilibrium DMD simulation.”

The behaviour of PS chains of different lengths in interaction with model cells is of great interest. PS chains can interact with cells, penetrating plasma membranes. Authors could complete the biography with recent papers reporting computational investigations on PS chains of similar length (G. Rossi et al., Polystyrene Nanoparticles Perturb Lipid Membranes, *J. Phys. Chem. Lett.* 2014, 5, 241–246; D. Bochicchio et al., Polystyrene perturbs the structure, dynamics, and mechanical properties of DPPC membranes: An experimental and computational study, *J. Colloid Interface Sci.* 605 (2022) 110–119).

A critical point is the direct comparison between in silico results on 20-mers PS and in vitro experimental results on PS commercial beads (30 nm).

Authors have to discuss this severe limitation to the study.

We agree with the reviewer that it is important to uncover cell interactions with PS of different lengths. Non-charged PS chains can interact with cells and penetrate plasma membranes as convincingly demonstrated by the referred computational and experimental works. Here, negatively charged PS modified with carboxyl groups was modelled as in the experiments to minimise direct interactions with cell membranes containing negatively charged lipids. We added a brief discussion on page 17 and the two suggested papers to the references.

Although the sizes of the PS and PMMA nanoplastics were not directly comparable to those in the experiments, due to the need for reducing the computational costs, additional simulations showed that a range of PS nanoplastics with increased size up to 80mer consistently disrupted the cadherin dimer. Hence, large nanoplastics are expected to behave similarly in interacting with VE-cadherin dimer. We added the new simulations results as Fig. S17 in the SI and a brief discussion on page 19.

4. “These regions were located away from the interfacial region of the dimer and were rich in positively charged residues, as indicated by the surface representation of the dimer color-coded according to the nanoplastic-binding frequencies. The binding frequency data suggested that the PS nanoplastic preferred to bind near the turn region of the dimer. Therefore, the electrostatic interactions between led to association of the two entities.” In silico results reveal that carboxylated PS chains interact with the dimer of VE cadherin through electrostatic interaction between the carboxyl groups on the surface of the PS nanoplastic and the positively charged residues in the turn region of the dimer, as reported in Figure 4. Authors must exclude that the interaction with VE cadherin is governed solely by the charge of the polymer, by carrying out tests both on negatively charged polymer chains other than plastics and on non-derivatized PS chains, to elucidate the intrinsic propensity of PS, if any, to interact directly with the dimer of VE cadherin leading to dimer dissociation.

As the reviewer pointed out, electrostatic interaction was the driving force for the binding between the negatively charged PS and the turn regions of VE-cadherin. Following the suggestions, we performed binding simulations of non-charged PS chain with VE-cadherin and found that the binding sites on VE-cadherin were different and the binding did not affect the dimer dissociation of VE-cadherin. We added the new data as Fig. S15 in the SI and a brief discussion on pages 17-18.

Reviewer #2 (Remarks to the Author):

The manuscript presents research of the potential for nanoplastics to induce endothelial leakiness. The hypothesis proposed has important implications for humans and other organisms with a vascular system and major significance given the pervasiveness of nanoplastics in the environment. The study is detailed, comprehensive and well documented. However, it completely relies on just one type (and size) of nanoplastic, which is not necessarily the most common in the environment and, as a purpose made plastic, may not be

representative of other environmental nanoplastics such as those produced by fragmentation and ageing of bulk plastics. This should at the very least be discussed in the paper and the claims made should be moderated accordingly.

We thank the reviewer for their strong endorsement of the importance of our study. As the most used model nanoplastic in literature (~98% of the publications we have surveyed), carboxylated PS nanoplastic possess good stability and water suspensibility, which guaranteed a higher repeatability of the experiments. Per the reviewer's suggestion, we have added measurements for two other types of nanoplastics, aminated PS (NH₂-PS) and poly(methyl methacrylate) (PMMA). Our transwell and confocal data (Figs. 2&3) have demonstrated that NanoEL could be induced by both negatively charged nanoplastics of carboxylated PS and PMMA, but not the positively charged nanoplastic (NH₂-PS). Specifically, we did not observe NanoEL with aminated-PS nanoplastic on HUVEC cells at different concentrations and time points, indicating that charge could modulate the interaction of PS nanoplastic with VE-cadherin, consistent with the findings of NanoEL with inorganic nanoparticles.¹

As shown in Fig. S16, we found that the PMMA nanoplastic had a similar strong binding to the same turn regions in cadherin as did PS nanoplastics, but also a weak binding to the interface regions (residues 1-4, 85-89) of the dimer. Subsequent sDMD simulations of the cadherin dimer with PMMA nanoplastic indicated that PMMA nanoplastic also increased cadherin dimer dissociation under low forces (Fig. S16d). Interestingly, due to the additional binding of PMMA to the cadherin dimer interface, an alternative dissociation pathway with competitive binding of the nanoplastic to one of the cadherin monomers was observed (Fig. S16e). Together with the experiments (Figs. 2-4), both anionic PS and PMMA nanoplastics have been shown to be NanoEL competent.

The manuscript is mostly well written but in places requires attention, as follows:

Line 83: replace “vines” with “veins”

Line 92: replace “mono-dispersive” with “monodisperse”

Lines 99-108: the discussion about ROS generation and cell death is written in a rather confusing way and requires revision.

Line 110: replace “microtuble” with “microtubule”

Lines 308-309: sentence starting “As the concentration...” needs revision

Line 351: replace “reveal” with “become apparent”

Lines 358-362: wording and claim made in this sentence needs revision (see also earlier discussion).

Lines 394-396: Sentence starting “Subsequently...” needs revision

Line 409: Sentence starting “Then cells...” needs revision

Thank you for the comments. We have modified the manuscript accordingly.

Reviewer #3 (Remarks to the Author):

NCOMMS-21-48035

Nanoplastic Exposure Induces Endothelial Leakiness

The manuscript from Wei et al. examines whether acute exposure to polystyrene nanoplastics (nPS) can disrupt the endothelial barrier, along with an assessment of the potential mechanisms of action.

‘Microplastics and nanoplastics exposure and effects’ is currently a very important research topic for researchers, policy-makers, regulators, industry, citizens, and society in general, since evidence indicate that humans are inevitably exposed, and that plastic particles have the potential for biodistribution and bioaccumulation inside the body to produce negative health consequences.

Most of this evidence has been obtained using ecotoxicity-related experimental models, therefore data from more relevant human experimental models such as the in vitro vascular endothelial system used here by the authors are at present very valuable. The hypothesis of nPS being able to alter the functionality of the vascular endothelium is interesting, novel, attractive, and supported by the fact that other particulate materials (nanomaterials) are known to cause this same effect. It is also relevant that nanoplastics can translocate through the gastrointestinal barrier to reach blood vessels. Thus, the vascular endothelium is an interesting secondary target. It is also of note that the health implications of nanoplastic causing endothelial leakiness are huge.

We thank the reviewer for your elaboration on the human health relevance of micro- and nanoplastic exposure and for strong endorsement of the importance of our study. We agree with these statements wholeheartedly.

Despite of the very high interest of the approach, the experimental data presented here is not convincing enough to recommend publication. Below are the main points that justify my decision:

-The doses of nPS are unusually high. Most of them fall in what is considered non-biologically relevant. For some of the endpoints, the authors use 1 milligram per milliliter. Even though there are not environmental/human estimates for nanoplastic exposure, it is very unlikely that these concentrations can reach the blood system. One could justify that using non-biological concentrations is still useful for risk assessment purposes, and that would at some point be acceptable, but the problem with nanoplastics (and nanomaterials in general) is that technical issues start to appear when using concentrations above 100 micrograms per milliliter: the material aggregates massively and deposit on the surface of the plate, covering the cells and preventing from normal cell growth and behavior. Any effect found under these conditions is irrelevant.

Thank you for this comment. NanoEL is a phenomenon originating from rupture of VE-cadherin junctions resulting from their single-molecular interactions with nanoparticles on the nanoscale and subsequent accumulation of such rupture on the microscopic scale to yield physical gaps in the paracellular space, as revealed by our in silico and signalling pathway data. For such a phenomenon to occur it only requires a finite small number of nanoparticles,

as shown by our in silico simulations, but to quantify such phenomenon accurately we had to use concentrations adequate for conventional instrumentation. We removed the data for 1 mg/mL, which was not used for inducing endothelial leakiness with HUVECs. We used concentrations above 50 μ g/mL for confocal microscopy and transwell experiments, as at lower nanoplastic concentrations NanoEL would not be clearly visible with conventional ensemble techniques, especially on confocal microscopy (resolution: \sim 0.2 μ m). On the other hand, the environmental concentrations of nanoplastics are highly sporadic as revealed by the literature in the past decade, and in biological systems accumulation of nanoparticles in tissues and organs, especially at biological barriers such as the paracellular space of endothelia, often has no linear or direct relationship with nanoparticle dose but is specific to time, location, tissue (due to nonlinear effects such as EPR, NanoEL) and nanoparticle physicochemical properties (such as hydrophobic association).

-Authors culture HUVECs in RPMI media + 10% FBS, which is not recommended. RPMI is low in cations. This may affect cell interactions that are cation-dependent, such as adherens junctions. Hence, endothelial leakiness could have been influenced by the culture media conditions.

Thank you for this comment. We have checked the effects of RPMI and endothelial cell media (ECM) on adherens junctions, which did not show statistically differences on confocal microscopy. In addition, we also included negative control at different time points in each assay to ensure the integrity of HUVEC monolayer and of adherens junctions as Figs.2c&3g.

-Authors poorly assess nPS internalization, localization in the cytosol, or the membrane fraction. The nPS used is fluorescent, hence a better assessment using confocal microscopy and flow cytometry is expected.

Also regarding nPS internalization, Figure S5 is presented in a way where time-dependent internalization cannot be evaluated.

Thank you for this comment. We have used flow cytometry to measure cell internationalisation of PS nanoplastic. The data has been shown as Fig. 1f.

-According to Figure 1- brightfield image, the concentration of 1 milligram per milliliter dose is causing extensive cell death (maybe due to technical issues explained in the previous point?), moreover not matching with Figure 1E.

Thank you for this comment. The calculation of cell mortality was based on the cells which still adhered onto the bottom of the well via software. The floating cells were washed away and were not included in the calculation. We have removed the data for 1 mg/mL as it was not used for inducing endothelial leakiness on HUVECs in the whole manuscript.

-nPS characterization should be evaluated in cell growth media (and not only miliQ water) to have a better estimate of the sizes and zeta potentials of the nPS in a media that better mimic the treatment conditions. Please indicate how many TEM images have been analyzed to determine the particle size.

Thank you for this suggestion. We have provided TEM and DLS data for PS nanoplastic in cell media in Figs. S1, S9, S10 and Table S1. We have analysed 5 TEM images to determine the particle size, including more than 300 PS nanoparticles.

-ROS data lacks positive control and time-matched controls. ROS is quite high at concentrations where no toxicity is found. Is this expected? The variability of ROS values seems low for this technique. A complementary technique to assess ROS is necessary.

Thank you for this comment. We have added a Cell Counting Kit-8(CCK8) assay to measure the cell proliferation/toxicity after their treatment with different nanoplastics. H₂O₂ with 200 μ M was used as positive control. The new data are shown in Figs. 1e&3c, d.

-Why have authors explored the mechanisms of nPS-induced cell death (autophagy and apoptosis) at time-points where cell death is not evident? Why they have not included the 22h where the maximum mortality rate is found?

Thanks for this question. Our study has identified the phenomenon of nPS-induced endothelial leakiness which occurred within a short period of time. During this time period, nPS did not induce significant toxicity or death to the endothelial cells. Here we intended to explore the mechanism further with autophagy and apoptosis assays, to further verify that nPS did not induce cell death at 1 h, when 0.5 mg/mL PS induced endothelial leakiness.

-Autophagy data lacks positive control and negative time-matched controls.

Thank you for the suggestion. We have added positive and negative time-matched controls to the autophagy data in Fig. 1g&h.

-The manuscript contains some contradictory statements when explaining whether authors use serum-free media for treatments or not, and the same for the concurrent controls. This needs to be clarified. Importantly, serum deprivation induces autophagy. Hence, reasonable doubts arise about nPS being the cause of autophagy induction.

Thanks for the suggestion. We have added negative time-matched controls to the autophagy data. Negative control group is medium without serum. At 3 h and 6 h, the level of autophagy in the nPS group was significantly higher than that in the negative control group. Thus nPS induced HUVECs autophagy at these time points.

-Authors conclude that nPS is not toxic under 0.5 mg/mL and 3 h of exposure. Based on what? Only cell mortality data? ROS and autophagy markers are positive at time-points and concentrations where they claim there is no toxicity. And we do not know about apoptosis because only 6h is shown.

Thanks for this question. There was no significant difference in cell viability for nPS exposure at 0.05 mg/mL and 0.5 mg/mL at 1 h, while the toxicity of nPS to cells was concentration- and time-dependent as shown in Fig. 1e. At 1 h after we observed leakiness (Fig. 2b), cell viability was not significantly reduced at 0.05 mg/mL and 0.5 mg/mL (Fig. 1e). Therefore, the toxicity

of nPS to cells was not the cause of leakiness, as also revealed by our signalling assays (Fig. 4). We have removed the statement that nPS was not toxic under 0.5 mg/mL and 3 h of exposure in the manuscript.

-Authors assess endothelial leakiness at time-points and concentrations where ROS and apoptosis are evident.

Thanks for this comment. The apoptosis assay was performed with 0.05 and 0.5 mg/mL of PS, and at 6 h-treatment. The transwell and confocal data showed endothelial leakiness on HUVEC cells at these points in Fig. 2c.

-It is not clear how authors assess that they have generated an intact monolayer in the transwell before applying the nPS treatment.

Thanks for this question. In each batch of the transwell assay, we performed the same seeding at 96 well plate and identical culture procedures to observe cell conditions over the time course. The well area of 96 well plate equalled to the area of the transwell insert. Once we observed monolayer in the 96 well plate, additional two days of culture were proceeded to ensure we had an intact monolayer in the transwell insert. After that, the nPS treatment were applied as planned.

-Negative control in Figure S6 should be included in the main Figure. Include also the rest of time-matched negative controls. Holes in the monolayer of the shown control are evident. Is this expected? The same for 1h of treatment in Figure 2C, although there are no arrows pointing to them. The quality of the negative control and the 1h of treatment pictures is lower than the rest. Overall, no solid data/images support the leakiness effect.

Thanks for this question. We have added the time-matched negative control in Fig. 1. In the non-treated control group, we obtained an intact HUVEC monolayer and very few holes could be found. Occasionally, we found very small holes in the control group under confocal due to the unavoidable movement of the cells.

To observe the phenomenon of endothelial leakiness on HUVECs induced by nPS, we have employed both quantitative and qualitative methods, such as the transwell assay and confocal microscopy. From the transwell data, endothelial leakiness induced by nPS was evidently confirmed. FITC-dextran could penetrate from the insert to the lower chamber due to the HUVEC monolayer leakiness induced by PS nanoplastic (Fig. 2b). Based on our confocal imaging, very few small holes could be observed in the control group compared with the positive control (EDTA, Fig. S7) and sample groups. In addition, the abundance of gaps was mapped and quantified (Fig. 2d) and the gap distribution images were also provided in Fig. S6&S7. The areas of the holes were calculated based on at least 3 images per sample.

-Why the authors didn't use a positive control for endothelial leakiness such as EDTA?

Thanks for this question. We have provided a confocal image for the positive control in Fig. S7, with EDTA at 0.5mM and 10 min treatment. The gap distribution derived from the image is also shown in Fig. S7.

-To assess the role of ROS and endocytosis, authors selected 1h of nPS treatment to assess endothelial leakiness in the transwell with and without the use of inhibitors or NAC (Figure 3). Why? According to Figure 2, no FITC-dextran was detected in the basolateral compartment at this condition.

Thanks for this question. The occurrence of NanoEL induced by inorganic nanoparticles has been demonstrated to occur in a very short period, typically within 30 to 60 min. In this study, we report on the endothelial leakiness associated with nanoplastic exposure. The incorporation of the endocytosis inhibitor NAG was to investigate whether the PS nanoplastic-induced endothelial leakiness was independent of ROS formation and endocytosis. Our data in Fig. 4 demonstrated that the NanoEL induced by the nanoplastic was related to VE-cadherin pathway and actin remodelling, rather than ROS and endocytosis.

Regarding the transwell assay, FITG-dextran was detectable in the basolateral compartment for all groups, including negative control. When we performed data analysis, we normalised the fluorescence intensity of negative control, which was set as 1. Hence, the Y axes indicated relative fluorescence intensities, not actual readings.

-Authors modeled a 1.5 nm nPS particle due to the computational cost. Is then the result comparable to nPs particles of 30 nm? This needs to be discussed/clarified.

We appreciate the reviewer's comment. Due to the high computational costs, we used a 1.5 nm 20-mer PS nanoplastic for effective investigation of their induced endothelial leakiness. This approach is commonly adopted for simulation studies to offer molecular insights to complement experiments. Following the comment, we varied the size of the PS nanoplastic to include 30, 40, 60, and 80-mers and assessed how different sizes of the PS nanoplastic impacted the cadherin dimer dissociation. As shown in Fig. S17, the sDMD simulations of the VE-cadherin dimer with different sizes of the nanoplastic were performed, and all different sized PS nanoplastic induced the dimer dissociation. Although our maximum size (80 mer) of PS nanoplastic is not comparable to the experimentally chosen 30 nm PS nanoplastic, we computationally identified that the increased size of PS nanoplastic could induce VE-cadherin dimer dissociation. Therefore, our overall results validated that the large sizes of PS nanoplastic could effectively induced the dimer dissociation.

-Indicate the equivalences of the in vivo selected dose.

Thank you for the suggestion. Mice in treatment groups were exposed to PS nanoplastic with the exposure doses of 1.5 mg/kg, 15 mg/kg and 30 mg/kg. The low dose was chosen according to the environmentally realistic concentration of PS in rivers². The middle dose was 10 times as much as the low dose. The high dose was 2 times as much as the middle dose. The exposure doses used here was based on the Precautionary Principle to reveal the target mechanisms first before evaluating the risk at environmentally relevant concentrations.

-Dose-dependency is not found in most of the organs after the in vivo EDB+nPS assay. This needs to be discussed.

Thanks for your comment. The living body is not an isolated system, therefore the concentrations of PS nanoplastic entering the tissue were much lower than the concentrations administered. Due to the small numbers of PS entering the tissue, a large portion PS nanoplastic was metabolized, so no dose-dependency was found in most of the organs after the in vivo EDB+nPS assay.

-A complementary in vivo analysis of direct vascular leakiness should be included to reinforce the findings i.e. inject nPS or vehicle into subcutaneous pockets followed by EBD injection.

Thanks for this comment. We have performed this experiment and the data are shown in Fig. 6e.

-Scheme in Figure S2A is not very helpful.

Thanks for your comment. We needed this figure to illustrate the related signalling pathway and up- and down-stream proteins, which we tested in the assays.

-Results and discussion section contain too many technical details that should be moved to methods or figure legends. Also, this section will benefit from including more statements regarding interpretation and discussion of the data.

Thanks for your comment. We have modified the manuscript accordingly to improve its readability and to give more details to our methods for enhanced clarity.

Reviewer #4 (Remarks to the Author):

This study investigated the potential toxicity of polystyrene nanoplastic (PS nanoplastic) to vascular endothelium. The results showed that vasculature permeability could be a new mechanism for the transport of nanoplastics. A series of experiments were conducted to investigate the underlying mechanisms, which revealed that exposure to PS nanoplastic induced phosphorylation of VE-cadherin, and further induced endothelial leakiness. The authors performed several validating experiments to make their conclusion reliable. However, there are a number of concerns. Revisions are required before consideration of publication. Specific comments are provided below.

We thank the reviewer for your endorsement of the importance of our study.

1. Carboxylated PS beads were chosen in this study. Is this the main form of polystyrene nanoplastic in nature?

Most studies in the literature (~98% of the publications we have surveyed) have used the commercially available carboxylated PS nanoplastic, which possess good stability and water susceptibility to ensure a higher repeatability of the experiments. In the revision, we have

added measurements for two other types of commercially available nanoplastics, aminated PS (NH₂-PS) and poly(methyl methacrylate) (PMMA). Our transwell and confocal data (Figs. 2&3) have demonstrated that NanoEL could be induced by both negatively charged nanoplastics of carboxylated PS and PMMA, but not by the positively charged nanoplastic (NH₂-PS).

2. Could all the nanoparticles (around 30 nm) contribute to this phenotype. In other words, Did the phenotype in this study is PS nanoplastic specifically induced?

Thank you for this question. We have employed another two types of commercially available nanoplastics, aminated polystyrene (NH₂-PS) and poly(methyl methacrylate) (PMMA) for the revision, to investigate how different types of nanoplastics may impact endothelial leakiness. The transwell and confocal data (Figs. 2&3) have demonstrated that, the phenotype of NanoEL could be induced by both negatively charged nanoplastics of carboxylated PS and PMMA, but not by the positively charged nanoplastic NH₂-PS. These results, together with our simulation results with PS and PMMA (Figs. 5, S14&S16), are consistent with NanoEL induced by differently charged inorganic nanoparticles¹.

3. p62 is a receptor for cargo destined to be degraded by autophagy.

Line 114-116, why both expression level of LC3-II and p62 were increased at 3 and 6 h? These results should be discussed.

Thanks for this question. The expression levels of LC3-II and p62 can reflect the strength of autophagy according to the publications³. When LC3-II was up-regulated and p62 was down-regulated, this suggested that autophagic flux in the cells was smooth. When LC3-II and p62 were both up-regulated, this revealed the dysfunction of autophagic degradation and autophagosome accumulation. When LC3-II and p62 were both up-regulated, it suggested that the autophagy progress was blocked and the autophagosomes and lysosomes were not fused. The expression level of LC3-II was increased at 3 and 6 h; meanwhile the expression level of p62 was increased at 6 h in our study, indicating that PS nanoplastic blocked autophagic flux.

4. Line 121-122: “We therefore concluded that PS nanoplastic under 0.5 mg/mL and within 3 h exposure would not cause significant toxicity ” . There several markers were changed in 3 hours, such as ROS, autophagy, etc. So more markers should be detected to support this conclusion.

Thanks for this suggestion. We used an additional CCK-8 assay to detect cell viability. We found that cell viability did not decrease significantly when PS nanoplastics were lower than 0.5 mg/mL and within 3 h exposure. It was therefore confirmed that PS nanoplastic under 0.5 mg/mL and within 3 h exposure did not induce significant toxicity.

5. More experiments should be performed to confirm that the PS nanoplastic really bound the VE-cadherin

Thanks for this suggestion. We performed an additional experiment to confirm that the PS nanoplastic bound VE-cadherin in Fig. 2f.

6. The phosphorylation of VE-cadherin should be also detected in the Ex vivo and In vivo model.

Thanks for your suggestion. For the expression of phosphorylated VE-cadherin in vivo model, it is tough to obtain mice vessels after administration of PS nanoplastic. So we settled for performing additional experiments with VE-cadherin phosphorylation ex vivo, and the data are shown in Fig. S13.

7. There are still studies conducted on rodents or on cell cultures that would be relevant to mention on the introduction and to consider as comparison on the discussion section:

<https://www.tandfonline.com/doi/abs/10.1080/17435390.2021.1930228?journalCode=inan20>

Thanks for your suggestion. We have cited this publication in our main text, as reference 9.

References

1. Tee, J. K., *et al.* Nanoparticles' interactions with vasculature in diseases. *Chem. Soc. Rev.* **48**, 5381-5407 (2019).
2. Eerkes-Medrano, D., Thompson, R. C., Aldridge, D. C. Microplastics in freshwater systems: A review of the emerging threats, identification of knowledge gaps and prioritisation of research needs. *Water Res.* **75**, 63-82 (2015).
3. Mohammadinejad, R., *et al.* Necrotic, apoptotic and autophagic cell fates triggered by nanoparticles. *Autophagy* **15**, 4-33 (2019).

REVIEWERS' COMMENTS

Reviewer #1 (Remarks to the Author):

Yang Song, Pu Chun Ke and coauthors submitted a revised version of the manuscript which includes not entirely exhaustive answers to the questions raised in the review process.

The main limit is that the new phenomenon, the disruption of vascular cadherin junction, seems dependent on the negative charge of nanomaterials with size < 100 nm and not on the nature of the material. The really interesting results should be obtained on neutral nanoplastics.

In the revised version authors added simulation results showing that unmodified PS chains (non-charged) bind to VE-cadherin, the binding site being different, and notably that the binding does not affect the dimer dissociation of VE-cadherin.

At least it is necessary to specify more clearly the limits of the results on VE-cadherin, perhaps even by changing the title.

Minor:

the discrepancy between the size of nanoparticles as determined by TEM and DLS is huge. What do you mean with hydration? Do you think that PS nanoparticles hydrodynamic size could be double the size observed by TEM, with a hydration shell larger than 10 nm? As you can see in the source files (figure S1, S9, S10) the intensity distributions are bimodal, showing that you have impurities or aggregates in your samples. You can either remove impurities to obtain good DLS data or report the bimodal distribution on the graphs.

Table S1. Authors should report only significant figures for size, PDI and Zeta-potential.

Reviewer #4 (Remarks to the Author):

my questions are well covered in the revised manuscript.

Response letter:

We thank the Reviewers for providing insightful comments on our revised manuscript. We have addressed these comments carefully in the second revision of our manuscript. Please find our point-by-point response below.

Reviewer #1 (Remarks to the Author):

Yang Song, Pu Chun Ke and coauthors submitted a revised version of the manuscript which includes not entirely exhaustive answers to the questions raised in the review process.

Thanks for this comment. For the first revision, we had tried our best to address the questions and requests for additional experiments and simulations made by the 4 reviewers, especially with the 2-month lockdowns in Shanghai, where some of the experiments were performed.

The main limit is that the new phenomenon, the disruption of vascular cadherin junction, seems dependent on the negative charge of nanomaterials with size < 100 nm and not on the nature of the material. The really interesting results should be obtained on neutral nanoplastics.

In the revised version authors added simulation results showing that unmodified PS chains (non-charged) bind to VE-cadherin, the binding site being different, and notably that the binding does not affect the dimer dissociation of VE-cadherin.

At least it is necessary to specify more clearly the limits of the results on VE-cadherin, perhaps even by changing the title.

Thanks for the valuable comments. According to the studies by Leong et al (Chem Soc Rev 2019, 48, 5381, and the relevant references therein), the NanoEL phenomenon has been known to occur for a range of anionic inorganic nanoparticles of given sizes (<100 nm) and densities (>1.72 g/m³). This current study has expanded that library to organic nanomaterials, i.e., nanoplastics of great relevance to environmental sustainability and human health. We showed in this study that anionic polystyrene and PMMA nanoplastics, despite their relatively low densities (~1.05 g/m³ for polystyrene and ~1.18 g/m³ for PMMA), were competent in disrupting VE-cadherins to create endothelial leakiness, thereby pointing to new environmental health and safety implications of plastics. Indeed, our additional simulation in the first revision showed binding of neutral polystyrene nanoplastic at different sites and incompetent disruption to VE-cadherin junction. In experiments, neutral plastics are typically not well suspendable in aqueous solutions unless they are afforded with surfactants, thus rendering experiments difficult to be repeatable. On the other hand, neutral plastics in the environment may gain surface charge due to the adsorption of organic matter and natural degradation (oxidation and weathering by sunlight and other factors). To avoid confusion and complication, we have modified the title of the manuscript to "Anionic Nanoplastic Exposure Induces Endothelial Leakiness".

Minor:

the discrepancy between the size of nanoparticles as determined by TEM and DLS is huge. What do you mean with hydration? Do you think that PS nanoparticles hydrodynamic size could be double the size observed by TEM, with a hydration shell larger than 10 nm? As you can see in the source files (figure S1, S9, S10) the intensity distributions are bimodal, showing that you have impurities or

aggregates in your samples. You can either remove impurities to obtain good DLS data or report the bimodal distribution on the graphs.

The differences in nanoplastic size between the TEM and DLS measurements indicated some degree of agglomeration, as is often the case with synthetic nanomaterials. Indeed, hydration was not an accurate term to describe such discrepancy. We have modified this statement on p4 in the second revision and updated the peak values in Supplementary Table 1.

Table S1. Authors should report only significant figures for size, PDI and Zeta-potential.

Thanks for your comment, we have updated the significant figures in Supplementary Table 1.

Reviewer #4 (Remarks to the Author):

my questions are well covered in the revised manuscript.

We thank the reviewer for their strong endorsement of our revised manuscript.